# RETRIEVE: Coreset Selection for Efficient and Robust Semi-Supervised Learning

**Krishnateja Killamsetty**    **Xujiang Zhao**    **Feng Chen**    **Rishabh Iyer**
Department of Computer Science
The University of Texas at Dallas
Richardson, Texas, USA
{krishnateja.killamsetty,xujiang.zhao,feng.chen,rishabh.iyer}@utdallas.edu

## Abstract

Semi-supervised learning (SSL) algorithms have had great success in recent years in limited labeled data regimes. However, the current state-of-the-art SSL algorithms are computationally expensive and entail significant compute time and energy requirements. This can prove to be a huge limitation for many smaller companies and academic groups. Our main insight is that training on a subset of unlabeled data instead of entire unlabeled data enables the current SSL algorithms to converge faster, significantly reducing computational costs. In this work, we propose RETRIEVE[1], a coreset selection framework for efficient and robust semi-supervised learning. RETRIEVE selects the coreset by solving a mixed discrete-continuous bi-level optimization problem such that the selected coreset minimizes the labeled set loss. We use a one-step gradient approximation and show that the discrete optimization problem is approximately submodular, enabling simple greedy algorithms to obtain the coreset. We empirically demonstrate on several real-world datasets that existing SSL algorithms like VAT, Mean-Teacher, FixMatch, when used with RETRIEVE, achieve a) faster training times, b) better performance when unlabeled data consists of Out-of-Distribution (OOD) data and imbalance. More specifically, we show that with minimal accuracy degradation, RETRIEVE achieves a speedup of around $3\times$ in the traditional SSL setting and achieves a speedup of $5\times$ compared to state-of-the-art (SOTA) robust SSL algorithms in the case of imbalance and OOD data. RETRIEVE is available as a part of the CORDS toolkit: https://github.com/decile-team/cords.

## 1   Introduction

Deep learning algorithms have had great success over the past few years, often achieving human or superhuman performance in various tasks like computer vision [10], speech recognition [18], natural language processing [5], and video games [45]. One of the significant factors attributing to the recent success of deep learning is the availability of large amounts of labeled data [55]. However, creating large labeled datasets is often time-consuming and expensive in terms of costs. Moreover, some domains like medical imaging require a domain expert for labeling, making it nearly impossible to create a large labeled set. In order to reduce the dependency on the availability of labeled data, semi-supervised learning (SSL) algorithms [7] were proposed to train models using large amounts of unlabeled data along with the available labeled data. Recent works [42, 56, 4, 53] show that semi-supervised learning algorithms can achieve similar performance to standard supervised learning using significantly fewer labeled data instances.

---

[1]co**R**esets for **E**fficien**T** and **R**obust sem**I**-sup**Er****V**ised l**E**arning

35th Conference on Neural Information Processing Systems (NeurIPS 2021).

However, the current SOTA SSL algorithms are compute-intensive with large training times. For example, from our personal experience, training a WideResNet model [60] on a CIFAR10 [27] dataset with 4000 labels using the SOTA FixMatch algorithm [53] for 500000 iterations takes around four days on a single RTX2080Ti GPU. This also implies increased energy consumption and an associated carbon footprint [54]. Furthermore, it is common to tune these

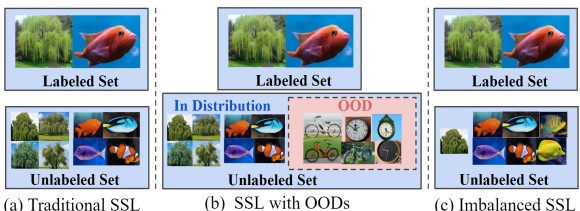

**(a) Traditional SSL**   **(b) SSL with OODs**   **(c) Imbalanced SSL**

Figure 1: (a )Unlabeled set with the same distribution as the labeled set, (b) Unlabeled set containing OOD instances, (c) Unlabeled set where the class distribution is imbalanced

SSL algorithms over a large set of hyper-parameters, which means that the training needs to be done hundreds and sometimes thousands of times. For example, [44] performed hyperparameter tuning by running 1000 trails of Gaussian Process-based Blackbox optimization[16] for each SSL algorithm (which runs for 500000 iterations). This process implies significantly higher experimental turnaround times, energy consumption, and CO2 emissions. Furthermore, this is not something that can be done at most universities and smaller companies. The first problem we try to address in this work is: *Can we efficiently train a semi-supervised learning model on coresets of unlabeled data to achieve faster convergence and reduction in training time?*

Despite demonstrating encouraging results on standard and clean datasets, current SSL algorithms perform poorly when OOD data or class imbalance is present in the unlabeled set [44, 8]. This performance degradation can be attributed to the fact that the current SSL algorithms assume that both the labeled set and unlabeled set are sampled from the same distribution. A visualization of OOD data and class imbalance in the unlabeled set is shown in Figure 1. Several recent works [59, 8, 17] were proposed to mitigate the effect of OOD in unlabeled data, in turn improving the performance of SSL algorithms. However, the current SOTA robust SSL method [17] is 3X slower than the standard SSL algorithms, further increasing the training times, energy costs, and CO2 emissions. The second problem we try to address in this work is: *In the case where OOD data or class imbalance exists in the unlabeled set, can we robustly train an SSL model on coresets of unlabeled data to achieve similar performance to existing robust SSL methods while being significantly faster?*

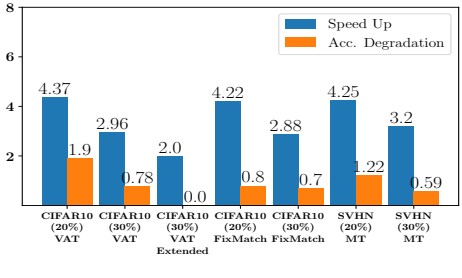
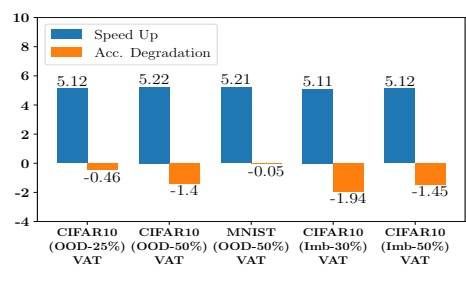

**(a) Traditional SSL**   **(b) Robust SSL**

Figure 2: Comparison of RETRIEVE with VAT, FixMatch, and MT on CIFAR-10 and SVHN: We contrast the accuracy degradation with speedup compared to the base SSL or robust SSL (DS3L) approach. We observe speedups of 3× in standard SSL case with 0.7% accuracy drop and 2× speedup with no accuracy drop. In the robust SSL case, we observe 5× speedup compared to DS3L [17] while outperforming it in terms of accuracy.

To this end, we propose RETRIEVE, a coreset selection framework that enables faster convergence and robust training of SSL algorithms. RETRIEVE selects coreset of the unlabeled data resulting in minimum labeled set loss when trained upon in a semi-supervised manner. Intuitively, RETRIEVE tries to achieve faster convergence by selecting data instances from the unlabeled set whose gradients are aligned with the labeled set gradients. Furthermore, RETRIEVE also achieves distribution matching by selecting a coreset from the unlabeled set with similar gradients to the labeled set.

## 1.1 Our Contributions

The contributions are our work can be summarized as follows:

- **RETRIEVE Framework:** We propose a coreset selection algorithm RETRIEVE for efficient and robust semi-supervised learning. RETRIEVE poses the coreset selection as a discrete-continuous bi-level optimization problem and solves it efficiently using an online approximation of single-step gradient updates. Essentially, RETRIEVE selects a coreset of the unlabeled set, which, when trained using the combination of the labeled set and the specific unlabeled data coreset, minimizes the model loss on the labeled dataset. We also discuss several implementation tricks to speed up the coreset selection step significantly (*c.f.*, Section 3.3, Section 3.4)

- **RETRIEVE in Traditional SSL:** We empirically demonstrate the effectiveness of RETRIEVE in conjunction with several SOTA SSL algorithms like VAT, Mean-Teacher, and FixMatch. The speedups obtained by RETRIEVE are shown in Figure 2a. Specifically, we see that RETRIEVE consistently achieves close to $3\times$ speedup with accuracy degradation of around $0.7\%$. RETRIEVE also achieves more than $4.2\times$ speedup with a slightly higher accuracy degradation. Furthermore, when RETRIEVE is trained for more iterations, RETRIEVE can match the performance of VAT while having a $2\times$ speedup (see VAT Extended bar plot in Figure 2a). RETRIEVE also consistently outperforms simple baselines like early stopping and random sampling.

- **RETRIEVE in Robust SSL:** We further demonstrate the utility of RETRIEVE for robust SSL in the presence of OOD data and imbalance in the unlabeled set. We observe that with the VAT SSL algorithm, RETRIEVE outperforms SOTA robust SSL method DS3L [17] (with VAT) while being around $5\times$ faster. RETRIEVE also significantly outperforms just VAT and random sampling.

## 1.2 Related Work

**Semi-supervised learning:** Several papers have been proposed for semi-supervised learning over the past few years. Due to space constraints, we do not talk about generative [50, 48, 26, 11, 19, 30, 3] and graph-based [62, 33] methods for SSL in this work. We instead focus on the main components of the existing SOTA SSL algorithms, viz., a) consistency regularization and b) entropy minimization. The consistency regularization component forces the model to have consistent prediction given an unlabeled data point and its perturbed (or augmented) version. The Entropy-minimization component forces the model instances to have low-entropy predictions on unlabeled data instances to ensure that the classes are well separated. One can achieve entropy minimization by directly adding the entropy loss component on the unlabeled class prediction or using methods like Pseudo-Labeling to enforce it implicitly. Mean-Teacher [56] approach uses a consistency regularization component that forces the predictions of the exponential moving average of the model to be the same as the model prediction of the augmented unlabeled images. VAT [42] instead computes the perturbation of the unlabeled data point that changes the prediction distribution the most and enforces the model to have the same prediction on both unlabeled data instance and unlabeled data instance with computed perturbation as a form of consistency regularization. MixMatch [4] uses $K$ standard image augmentations for consistency regularization and enforces entropy minimization by using a sharpening function on the average predicted distribution of $K$ augmentations of unlabeled data instances. FixMatch [53] induces consistency regularization by forcing the model to have the same prediction on a weakly augmented and strongly augmented image instance. Furthermore, FixMatch [53] also employs confidence thresholding to mask unlabeled data instances on which the model's prediction confidence is below a threshold from being used in consistency loss.

**Robust Semi-supervised learning:** Several methods have been proposed to make the existing semi-supervised learning algorithms robust to label noise in labeled data and robust to OOD data in the unlabeled set. A popular approach [49, 52] to deal with label noises and class imbalance in a supervised learning setting is by reweighing each data instance and jointly learning these weights along with the model parameters. Safe-SSL (DS3L) [17] is a recently proposed SOTA method for robust SSL learning. DS3L is similar to the reweighting in the supervised case and adopts a reweighting approach to deal with OOD data in the unlabeled set. Safe-SSL uses a neural network to predict the weight parameters of unlabeled instances that result in maximum labeled set performance, making it a bi-level optimization problem. In this regard, both RETRIEVE and Safe-SSL approach solves a bi-level optimization problem, except that RETRIEVE solves a discrete optimization problem at the outer level, thereby enabling significant speedup compared to SSL algorithms and an even more considerable

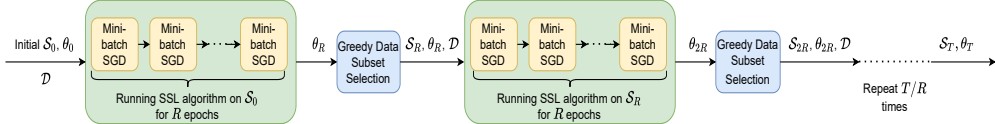

Figure 3: Flowchart of RETRIEVE framework, where coreset selection is performed every $R$ epochs and the model is trained on the selected coreset.

speedup compared to safe-SSL (which itself is $3\times$ slower than SSL algorithms). In contrast to safe-SSL and other robust SSL approaches, RETRIEVE achieves both efficiency and robustness. Other approaches for robust SSL include UASD [9] uses an Uncertainty aware self-distillation with OOD filtering to achieve robust performance and a distributionally robust model to deal with OOD [8].

**Coreset and subset selection methods:** Coresets [13] are small and informative weighted data subsets that approximate original data. Several works [57, 39, 24, 23] have studied coresets for efficient training of deep learning models in the supervised learning scenarios. CRAIG [39] selects representative coresets of the training data that closely estimates the full training gradient. Another approach, GLISTER [24] posed the coreset selection as optimizing the validation set loss for efficient learning focused on generalization. Another approach, GRAD-MATCH [23] select subsets that approximately match the full training loss or validation loss gradient using orthogonal matching pursuit. Similarly, coreset selection methods [57, 51, 2, 24] were also used for active learning scenario, where a subset of data instances from the unlabeled set is selected to be labeled. Finally, several recent works have used submodular functions for finding diverse and representative subsets for data subset selection [34, 21, 57, 58].

## 2   Preliminaries

**Notation:** Denote $\mathcal{D} = \{x_i, y_i\}_{i=1}^n$ to be the labeled set with $n$ labeled data points, and $\mathcal{U} = \{x_j\}_{j=1}^m$ to be the unlabeled set with $m$ data points. Let $\theta$ be the classifier model parameters, $l_s$ be the labeled set loss function (such as cross-entropy loss) and $l_u$ be the unlabeled set loss, e.g. consistency-regularization loss, entropy loss, etc.. Denote $L_S(\mathcal{D}, \theta) = \sum_{i \in \mathcal{D}} l_s(\theta, x_i, y_i)$ and $L_U(\mathcal{U}, \theta, \boldsymbol{m}) = \sum_{j \in \mathcal{U}} \boldsymbol{m}_i l_u(x_j, \theta)$ where $\boldsymbol{m} \in \{0,1\}^m$ is the binary mask vector for unlabeled set. For notational convenience, we denote $l_{si}(\theta) = l_s(x_i, y_i, \theta)$ and denote $l_{uj}(\theta) = l_u(x_j, \theta)$.

**Semi-supervised loss:** Following the above notations, the loss function for many existing SSL algorithms can be written as $L_S(\mathcal{D}, \theta) + \lambda L_U(\mathcal{U}, \theta, \boldsymbol{m})$, where $\lambda$ is the regularization coefficient for the unlabeled set loss. For Mean Teacher [56], VAT[42], MixMatch [4], the mask vector $\boldsymbol{m}$ is made up entirely of ones, whereas for FixMatch [53], $\boldsymbol{m}$ is confidence-thresholded binary vector, indicating whether to include an unlabeled data instance or not. Usually, $L_S$ is a cross-entropy loss for classification experiments and squared loss for regression experiments. A detailed formulation of the loss function $L_U$ used in different SSL algorithms is given in Appendix C

**Robust Semi-supervised loss:** However, for robust semi-supervised loss, the mask vector $\boldsymbol{m}$ is replaced with a weight vector $\boldsymbol{w} \in \mathbb{R}^m$ denoting the contribution of data instances in the unlabeled set. The weight vector $\boldsymbol{w}$ is unknown and needs to be learned. The weighted SSL loss is: $L_S(\mathcal{D}, \theta) + \lambda L_U(\mathcal{U}, \theta, \boldsymbol{w})$, where $\lambda$ is the regularization coefficient for the unlabeled set loss.

The state-of-the-art robust SSL method, Safe-SSL [17] poses the learning problem as:

$$\boldsymbol{w}^* = \underset{\boldsymbol{w}}{\text{argmin}} \, L_S(\mathcal{D}, \overbrace{\underset{\theta}{\text{argmin}} \, \underbrace{(L_S(\mathcal{D}, \theta) + \lambda L_U(\mathcal{U}, \theta, \boldsymbol{w}))}_{inner-level}}^{outer-level}) \tag{1}$$

In order to solve the problem at the inner level efficiently, Safe-SSL [17] method uses a single-gradient step approximation to estimate the inner problem solution. The weight vector learning problem after the one-step approximation is: $\boldsymbol{w}^* = \underset{\boldsymbol{w}}{\text{argmin}} \, L_S(\mathcal{D}, \theta - \alpha \nabla_\theta L_S(\mathcal{D}, \theta) - \alpha \lambda \nabla_\theta L_U(\mathcal{U}, \theta, \boldsymbol{w})$

Safe-SSL also uses a single-step gradient approximation to solve the outer level problem as well. As discussed before, the optimization problem of the Safe-SSL [17] algorithm involves continuous optimization at both inner and outer levels, whereas for RETRIEVE, the outer level involves a discrete optimization problem which makes it significantly faster than Safe-SSL.

## 3 RETRIEVE framework

In RETRIEVE, the coreset selection and classifier model learning on the selected coreset is performed in conjunction. As shown in the Figure 3, RETRIEVE trains the classifier model on the previously selected coreset for $R$ epochs in a semi-supervised manner, and every $R^{th}$ epoch, a new coreset is selected, and the process is repeated until the classifier model reaches convergence, or the required number of epochs is reached. *The vital feature of* RETRIEVE *is that the coresets selected are adapted with the training.* Let $\theta_t$ be the classifier model parameters and the $\mathcal{S}_t$ be the coreset at time step $t$. Since coreset selection is done every $R$ epochs, we have $\mathcal{S}_t = \mathcal{S}_{\lfloor t/R \rfloor}$, or in other words, the subsets change only after $R$ epochs. The SSL loss function on the selected coreset at iteration $t$ is as follows:

$$L_S(\mathcal{D}, \theta_t) + \lambda_t \sum_{j \in \mathcal{S}_t} \boldsymbol{m}_{jt} l_u(x_j, \theta_t) \tag{2}$$

where $\boldsymbol{m}_{jt}$ is the mask binary value associated with the $j^{th}$ point based on model parameters $\theta_t$ and $\lambda_t$ is the unlabeled loss coefficient at iteration $t$. Note that objective function given in Equation (2) is dependent on the SSL algorithm used in RETRIEVE framework. If gradient descent is used for learning, the parameter update step from time step $t$ to $t+1$ is as follows:

$$\theta_{t+1} = \theta_t - \alpha_t \nabla_\theta L_S(\mathcal{D}, \theta_t) - \alpha_t \lambda_t \sum_{j \in \mathcal{S}_t} \boldsymbol{m}_{jt} \nabla_\theta l_u(x_j, \theta_t) \tag{3}$$

where $\alpha_t$ is the learning rate at iteration $t$. The update step for mini-batch SGD is similar, just that it does the above on minibatches of the dataset.

### 3.1 Problem Formulation

The coreset selection problem of RETRIEVE at timestep $t$ is as follows:

$$\mathcal{S}_t = \overbrace{\underset{\mathcal{S} \subseteq \mathcal{U}: |\mathcal{S}| \leq k}{\operatorname{argmin}} L_S \Big( \mathcal{D}, \underbrace{\underset{\theta}{\operatorname{argmin}} \big( L_S(\mathcal{D}, \theta_t) + \lambda_t \sum_{j \in \mathcal{S}} \boldsymbol{m}_{jt} l_u(x_j, \theta_t) \big)}_{inner-level} \Big)}^{outer-level} \tag{4}$$

where $k$ is the size of the coreset and $\boldsymbol{m}_{jt}$ is the binary value associated with the $j^{th}$ instance based on model parameters $\theta_t$. $k$ is a fraction of the entire dataset (e.g. 20% or 30%), and the goal is to select the best subset of the unlabeled set, which maximizes the labeled loss based. The outer level of the above optimization problem is a discrete subset selection problem. However, solving the inner-optimization problem naively is computationally intractable, and, we need to make some approximations.

### 3.2 One-Step Gradient Approximation

To solve the inner optimization problem efficiently, RETRIEVE adopts a one-step gradient approximation based optimization method similar to [14, 49]. More specifically, RETRIEVE approximates the solution to the inner level problem by taking a single gradient step towards the descent direction of the loss function. The idea here is to jointly optimize the model parameters and the subset as the learning proceeds. After this approximation, the coreset selection optimization problem becomes:

$$\mathcal{S}_t = \underset{\mathcal{S} \subseteq \mathcal{U}: |\mathcal{S}| \leq k}{\operatorname{argmin}} L_S(\mathcal{D}, \theta_t - \alpha_t \nabla_\theta L_S(\mathcal{D}, \theta_t) - \alpha_t \lambda_t \sum_{j \in \mathcal{S}} \boldsymbol{m}_{jt} \nabla_\theta l_u(x_j, \theta_t)) \tag{5}$$

However, even after this approximation, the above optimization problem (Equation (5)) is NP-hard.

**Theorem 1** *Optimization problem (Equation (5)) is NP hard, even if $l_s$ is a convex loss function. If the labeled set loss function $l_s$ is cross-entropy loss, then the optimization problem give in the Equation (5) can be converted into an instance of cardinality constrained weakly submodular maximization.*

The proof is given in Appendix B. The given Theorem 1 holds as long as $l_s$ is a cross-entropy loss irrespective of the form of $l_u$. Further, Theorem 1 implies that the optimization problem given in Equation (5) can be solved efficiently using greedy algorithms [37, 38] with approximation guarantees. RETRIEVE uses stochastic-greedy algorithm [22, 38] to solve the optimization problem Equation (5) with an approximation guarantee of $1 - 1/e^{\beta} - \epsilon$ in $\mathcal{O}(m \log(1/\epsilon))$ iterations where $m$ is the unlabeled set size and $\beta$ is the weak submodularity coefficient (see Appendix B). And the set function used in stochastic greedy algorithm is as follows:

$$f(\theta_t, \mathcal{S}) = -L_S(\mathcal{D}, \theta_t - \alpha_t \nabla_\theta L_S(\mathcal{D}, \theta_t) - \alpha_t \lambda_t \sum_{j \in \mathcal{S}} \boldsymbol{m}_{jt} \nabla_\theta l_u(x_j, \theta_t)) \tag{6}$$

Notice that during each greedy iteration, we need to compute the set function value $f(\theta_t, \mathcal{S} \cup e)$ to find the maximal gain element $e$ that can be added to the set $\mathcal{S}$. This implies that the loss over the entire labeled set needs to be computed multiple times for each greedy iteration, making the entire greedy selection algorithm computationally expensive.

### 3.3 RETRIEVE Algorithm

To make the greedy selection algorithm efficient, we approximate the set function value $f(\theta_t, \mathcal{S} \cup e)$ with the first two terms of it's Taylor-series expansion Let, $\theta^S = \theta_t - \alpha_t \nabla_\theta L_S(\mathcal{D}, \theta_t) - \alpha_t \lambda_t \sum_{j \in \mathcal{S}} \boldsymbol{m}_{jt} \nabla_\theta l_u(x_j, \theta_t)$. The modified set function value with Taylor-series approximation is as follows:

$$\hat{f}(\theta_t, \mathcal{S} \cup e) = -L_S(\mathcal{D}, \theta^S) + \alpha_t \lambda_t \nabla_\theta L_S(\mathcal{D}, \theta^S)^T \boldsymbol{m}_{et} \nabla_\theta l_u(x_e, \theta_t) \tag{7}$$

where $\boldsymbol{m}_{et}$ is the binary mask value associated with element $e$. Note that the term $\boldsymbol{m}_{et} \nabla_\theta l_u(x_e, \theta_t)$ can be precomputed at the start of the greedy selection algorithm, and the term $\nabla_\theta L_S(\mathcal{D}, \theta^S)$ needs to be computed only once every greedy iteration, thereby reducing the computational complexity of the greedy algorithm.

A detailed pseudo-code of the RETRIEVE algorithm is given in Algorithm 1. RETRIEVE uses a greedy selection algorithm for coreset selection, and the detailed pseudo-code of the greedy algorithm is given in Algorithm 2. RETRIEVE can be easily implemented with popular deep learning frameworks [47, 1] that provide auto differentiation functionalities. In all our experiments, we set $R = 20$, i.e., we update the coreset every 20 epochs.

### 3.4 Additional Implementation Details:

In this subsection, we discuss additional implementational and practical tricks to make RETRIEVE scalable and efficient.

**Last-layer gradients.** Computing the gradients over deep models is time-consuming due to an enormous number of parameters in the model. To address this issue, we adopt a last-layer gradient approximation similar to [2, 39, 24, 23] by

---

**Algorithm 1: RETRIEVE Algorithm**

**Input:** Labeled Set: $\mathcal{D}$, Unlabeled Set: $\mathcal{U}$, Reg. Coefficients: $\{\lambda_t\}_{t=0}^{t=T-1}$, Learning rates: $\{\alpha_t\}_{t=0}^{t=T-1}$
**Input:** Total no of epochs: $T$, Epoch interval for subset selection: $R$, Size of the coreset: $k$

Set $t = 0$; Randomly initialize model parameters $\theta_0$ and coreset $\mathcal{S}_0 \subseteq \mathcal{U} : |\mathcal{S}_0| = k$;
**repeat**
  **if** $(t\%R == 0) \wedge (t > 0)$ **then**
    $\mathcal{S}_t = \text{GreedySelection}(\mathcal{D}, \mathcal{U}, \theta_t, \lambda_t, \alpha_t, k)$
  **else**
    $\mathcal{S}_t = \mathcal{S}_{t-1}$
  Compute batches $\mathcal{D}_b = ((x_b, y_b); b \in (1 \cdots B))$ from $\mathcal{D}$
  Compute batches $\mathcal{S}_{tb} = ((x_b); b \in (1 \cdots B))$ from $\mathcal{S}$
  *** Mini-batch SGD ***
  Set $\theta_{t0} = \theta_t$
  **for** $b = 1$ *to* $B$ **do**
    Compute mask $\boldsymbol{m}_t$ on $\mathcal{S}_{tb}$ from current model parameters $\theta_{t(b-1)}$
    $\theta_{tb} = \theta_{t(b-1)} - \alpha_t \nabla_\theta L_S(\mathcal{D}_b, \theta_t) - \alpha_t \lambda_t \sum_{j \in \mathcal{S}_{tb}} \boldsymbol{m}_{jt} \nabla_\theta l_u(x_j, \theta_t(b-1))$
  Set $\theta_{t+1} = \theta_{tB}$
  $t = t + 1$
**until** $t \geq T$
**return** $\theta_T, \mathcal{S}_T$

---

only considering the last classification layer gradients of the classifier model in RETRIEVE. By simply using the last-layer gradients, we achieve significant speedups in RETRIEVE.

**Warm-starting data selection:** We warm start the classifier model by training it on the entire unlabeled dataset for a few epochs similar to [23]. Warm starting allows the classifier model to have a

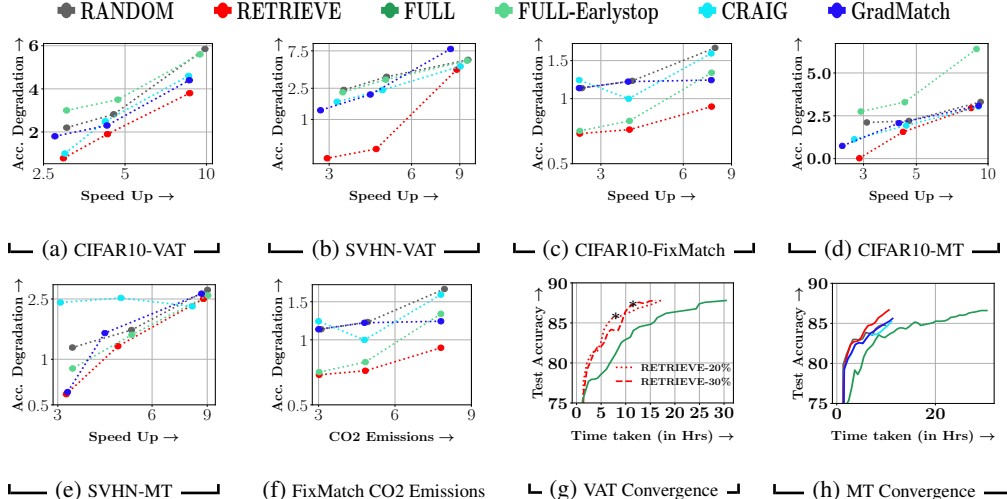

Figure 4: A comparison of RETRIEVE with baselines (RANDOM, CRAIG, FULL, and FULLEARLYSTOP in the traditional SSL setting. SpeedUp vs Accuracy Degradation, both compared to SSL for (a) VAT on CIFAR-10, (b) VAT on SVHN, (c) FixMatch on CIFAR-10, (d) MT on CIFAR-10, and (e) MT on SVHN. We observe that RETRIEVE significantly outperforms existing baselines in terms of accuracy degradation and speedup tradeoff compared to SSL. Plot (f) also compares the CO2 emissions among the different approaches on FixMatch, showing again that RETRIEVE achieves the best energy-accuracy tradeoff. Plots (g) and (h) are the convergence results comparing accuracy with the time taken. Again, we see that RETRIEVE achieves much faster convergence than all baselines and full training.

good starting point to provide informative loss gradients used for coreset selection. More specifically, we train the classifier model on the entire unlabeled set for $T_w = \frac{\kappa T k}{m}$ epochs where $k$ is coreset size, $T$ is the total number of epochs, $\kappa$ is the fraction of warm start, and $m$ is the size of the unlabeled set. To be fair, we consider all baselines in the standard SSL setting with the same warm start.

## 3.5 Epochs vs. Iterations

Most of the existing SSL algorithms are trained using a fixed number of iterations instead of epochs. However, for easier comprehension of the RETRIEVE algorithm, we use the epoch notation in our work. A single epoch here meant a pass over random mini-batches of data points, such that the total number of data points encountered is equal to the size of the coreset of the unlabeled data. For example, if the unlabeled set size is 50000 and the unlabeled batch size is 50, then a single epoch over 100%, 50%, and 30% subsets are equivalent to 1000, 500, and 300 iterations, respectively.

## 4 Experiments

Our experimental section aims to verify the efficiency and effectiveness of RETRIEVE by evalu-

---

**Algorithm 2:** GreedySelection

**Input:** Labeled Set: $\mathcal{D}$, Unlabeled Set: $\mathcal{U}$, Model Parameters: $\theta_t$, Learning rate: $\alpha_t$
**Input:** Regularization Coefficient: $\lambda_t$, Budget: $k$
Initialize $\mathcal{S}_t = \emptyset$
Set $m = |\mathcal{U}|$
Compute mask values $\boldsymbol{m}$ based on current model parameters $\theta_t$
**for** $e \in \mathcal{U}$ **do**
$\quad$ Compute $\theta_e = \boldsymbol{m}_e \nabla_\theta l_u(x_e, \theta_t)$

**for** $i = 1$ *to* $k$ **do**
$\quad$ Compute $\theta^S =$
$\quad\quad \theta_t - \alpha_t \nabla_\theta L_S(\mathcal{D}, \theta_t) - \alpha_t \lambda_t \sum_{j \in \mathcal{S}} \boldsymbol{m}_{jt} \nabla_\theta l_u(x_j, \theta_t)$

$\quad$ Compute $\nabla_\theta L_S(\mathcal{D}, \theta^S)$
$\quad$ $V \sim$ Sample $\lceil m \log(1/\epsilon) \rceil$ instances from $\mathcal{U}$
$\quad$ $best - gain = -\infty$ **for** $e \in V$ **do**
$\quad\quad$ Compute gain $\hat{g}(e) =$
$\quad\quad\quad \alpha_t \lambda_t \nabla_\theta L_S(\mathcal{D}, \theta^S)^T \boldsymbol{m}_e \nabla_\theta l_u(x_e, \theta_t)$
$\quad\quad$ **if** $\hat{g}(e) > best - gain$ **then**
$\quad\quad\quad$ Set $s = e$
$\quad\quad\quad$ Set $best - gain = \hat{g}(e)$

$\quad$ $\mathcal{S}_t = \mathcal{S}_t \cup s$
$\quad$ $\mathcal{U} = \mathcal{U} \setminus s$
**return** $\mathcal{S}_t$

---

ating RETRIEVE through three semi-supervised learning scenarios a) traditional SSL scenario with clean data, b) robust SSL with OOD, and c) robust SSL with class imbalance, to demonstrate the efficiency and the robustness of RETRIEVE. Furthermore, our work's experimental scenarios are very relevant in terms of research and real-world applications. We have implemented the RETRIEVE algorithmic framework using PyTorch [46]. We repeat the same experiment for three runs with different initialization and report the mean test accuracies in our plots. A detailed table with both

mean test accuracy and the standard deviations was given in Appendix (G, H). For a fair comparison, we use the same random seed in each trial for all methods. We explain implementation details, datasets, and baselines used in each scenario in the following subsections.

**Baselines in each setting.** In this section, we discuss baselines that are used in all the scenarios considered. We begin with the **traditional SSL scenario**. In this setting, we run RETRIEVE (and all baselines) with warm-start. We incorporate RETRIEVE with three representative SSL methods, including Mean Teacher (MT) [56], Virtual Adversarial Training (VAT) [42] and FixMatch [53]. The baselines considered are RANDOM (where we just randomly select a subset of unlabeled data points of the same size as RETRIEVE), CRAIG [39, 23] and FULL-EARLYSTOP. CRAIG [39, 23] was actually proposed in the supervised learning scenario. We adapt it to SSL by choosing a representative subset of unlabeled points such that the gradients are similar to the unlabeled loss gradients. We run the per-batch variant of CRAIG proposed in [23], where we select a subset of mini-batches instead of data instances for efficiency and scalability. Similarly, we use the per-batch version of GRADMATCH proposed in [23] adapted to SSL setting as another baseline. For more information on the formulation of CRAIG and GRADMATCH in the SSL case, see Appendix D, E. Again, we emphasize that RANDOM, CRAIG, and GRADMATCH are run with early stopping for the same duration as RETRIEVE. In FULL-EARLYSTOP baseline, we train the model on the entire unlabeled set for the time taken by RETRIEVE and report the test accuracy. In the traditional SSL scenario, we use warm variants of RETRIEVE, RANDOM, CRAIG for SSL training because warm variants are better in accuracy and efficiency compared to not performing warm start – see Appendix G for a careful comparison of both. **Robust SSL with OOD and Imbalance:** In the robust learning scenario for both OOD and imbalance, we analyze the performance of RETRIEVE with the VAT [42] algorithm. Note that for the Robust SSL scenario, we *do not warm start the model by training for a few iterations on the full unlabeled set* because training on an entire unlabeled set(containing OOD or class imbalance) creates a biased model due to a distribution mismatch between labeled set and unlabeled set. We empirically compare not warm starting the model with warm starting in Appendix H. In the robust SSL case, we compare RETRIEVE with two robust SSL algorithms DS3L [17] and L2RW [49]. DS3L (also called Safe-SSL) is a robust learning approach using a meta-weight network proposed specifically for robust SSL. We adapt L2RW(Learning to Reweight), originally proposed for robust supervised learning, to the SSL case and use it as a baseline. Similarly, we adapt the robust coreset selection method CRUST [40] originally proposed to tackle noisy labels scenario in supervised learning to SSL setting and use it as a baseline in Robust SSL scenario.

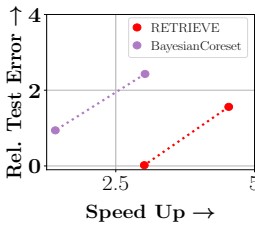

Figure 5: Performance comparison of RETRIEVE vs Bayesian Coreset using VAT for 20% and 30% CIFAR10 subsets

**Datasets, Model architecture and Experimental Setup:** We begin by providing details common across the three scenarios. We perform experiments on the following image classification datasets: CIFAR-10 [28] (60000 instances), SVHN [43] (99289 instances) and the following sentiment analysis datasets: IMDB [36][2] (10000 instances), and ELEC [20][3](246714 instances) datasets. We use a modified version of the ELEC dataset where the duplicate sentences are removed. For CIFAR-10, we use a labeled set of 4000 instances with 400 instances from each class, an unlabeled set of 50000 instances, a test set of 10000 instances. For SVHN, we used a labeled set of 1000 instances with 100 instances from each class, an unlabeled set of 73257 instances, a test set of 26032 instances. For IMDB and ELEC datasets, we use the labeled and unlabeled splits following the work [41]. For CIFAR10 and SVHN datasets, we use the Wide-ResNet-28-2 [60] model that is commonly used in SSL [44, 53]. For MNIST, we use a variant of LeNet [31] (see Appendix F for details). For IMDB and ELEC datasets, we use a model comprising of Word Embedding layer, LSTM model, and two-layer MLP model following the architecture given in the work [41]. Similar to the work [41], we initialize the embedding matrix and LSTM model weights using a pretrained recurrent language model using both the labeled and unlabeled data. For Image datasets, with RETRIEVE(and baselines like RANDOM, GRADMATCH and CRAIG), we use the Nesterov's accelerated SGD optimizer with a learning rate of 0.03, weight decay of 5e-4, the momentum of 0.9, and a cosine annealing [35] learning rate scheduler for all the experiments. For the FULL and FULLEARLYSTOP, we use the Adam optimizer [25] and follow the experimental setting from the SSL papers [53, 42, 56].

---

[2]https://ai.stanford.edu/ amaas/data/sentiment/

[3]http://riejohnson.com/cnn_data.html

For Text datasets, with RETRIEVE(and baselines like RANDOM, GRADMATCH and CRAIG), we set the hyperparameter values following the work [41]. Similarly, for the robust SSL baselines, we use the settings from the corresponding papers [17]. For all our experiments using image datasets, we use a batch size of 50 for labeled and unlabeled sets. Next, we discuss the specific settings for the **traditional SSL scenario**. For CIFAR10, we train the model for 500 epochs, and for SVHN, we train the model for 340 epochs on an unlabeled set. Note that we mention the epochs here because the number of iterations depends on the size of the unlabeled sets since that would determine the number of mini-batches. For a fair comparison, we train all algorithms for a fixed number of epochs. Next, we look at **robust SSL for OOD**. In this scenario, we consider the presence of OOD in the unlabeled set. We introduce OOD into CIFAR-10 following [44], by adapting it to a 6-class dataset, with 400 labels per class (from the 6 animal classes) as ID and rest of the classes as OOD (ID classes are: "bird", "cat", "deer", "dog", "frog", "horse", and OOD data are from classes: "airline", "automobile", "ship", "truck"). Similarly, we adapt MNIST [32] to a 6-class dataset, with classes 1-6 as ID and classes 7-10 as OOD. We denote the OOD ratio=$\mathcal{U}_{ood}/(\mathcal{U}_{ood} + \mathcal{U}_{in})$ where $\mathcal{U}_{in}$ is ID unlabeled set, $\mathcal{U}_{ood}$ is OOD unlabeled set. For CIFAR10, we use a labeled set of 2400 instances and an unlabeled set of 20000 instances, and for MNIST, we use a labeled set of 60 instances and an unlabeled set of 30000 instances. Finally, for **robust SSL for class imbalance**, we consider imbalance both in the labeled set and unlabeled set. We introduce imbalance into the CIFAR-10 dataset by considering classes 1-5 as imbalanced classes and a class imbalance ratio. The class imbalance ratio is defined as the ratio of instances from classes 1-5 and the number of instances from classes 6-10. For CIFAR-10, we use a labeled set of 2400 and an unlabeled set of 20000 instances.

**Traditional SSL Results:** The results comparing the accuracy-efficiency tradeoff between the different subset selection approaches are shown in Figure 4. We compare the performance for different subset sizes of the unlabeled data: 10%, 20%, and 30% and three representative SSL algorithms VAT, Mean-Teacher, and FixMatch. For warm-start, we set kappa value to $\kappa = 0.5$ (i.e., training for 50% epochs on the entire unlabeled set and 50% using coresets). Our experiments use a $R$ value of 20 (i.e., coreset selection every 20 epochs). Sub-figures(4a, 4b, 4d, 4e, 4c) shows the plots of relative error vs speedup, both w.r.t full training (i.e., original SSL algorithm).

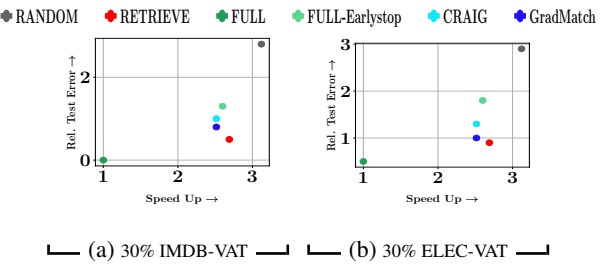

Figure 6: A comparison of RETRIEVE with baselines (RANDOM, CRAIG, GRADMATCH, FULL, and FULLEARLYSTOP in the traditional SSL setting for text datasets. SpeedUp vs Accuracy Degradation, both compared to SSL for (a) VAT on 30% IMDB, (b) VAT on 30% ELEC.

Sub-figure 4f shows the plot of relative error vs CO2 emissions efficiency, both w.r.t full training. CO2 emissions were estimated based on the total compute time using the Machine Learning Impact calculator presented in [29]. From the results, it is evident that RETRIEVE achieved the best speedup vs. accuracy tradeoff and is environmentally friendly based on CO2 emissions compared to other baselines (including CRAIG and FULLEARLYSTOP). In particular, RETRIEVE achieves speedup gains of 2.7x and 4.4x with a performance loss of 0.7% and 0.3% using VAT on CIFAR10 and SVHN datasets. Further, RETRIEVE achieves speedup gains of 2.9x, 3.2x with a performance loss of 0.02% and 0.5% using Mean-Teacher on CIFAR10 and SVHN datasets. Additionally, RETRIEVE achieves a speedup of 3.8x with a performance loss of 0.7% using FixMatch on the CIFAR10 dataset. Sub-figures(6a, 6b) shows the plots of relative error vs speedup both w.r.t full training (i.e., original SSL algorithm) on IMDB and ELEC datasets for 30% subset size. In particular, RETRIEVE achieves speedup gains of 2.68x and 2.5x with a performance loss of 0.5% and 0.6% for 30% subset of IMDB and ELEC datasets. Figure 5 shows the results comparing the BAYESIANCORESET method [6], adapted to the SSL setting with RETRIEVE using the VAT algorithm for 20% and 30% CIFAR10 subsets. The results show that RETRIEVE achieves better performance than the SSL extension of the BAYESIANCORESET selection method in terms of model performance and speedup. One possible explanation for it is that the BAYESIANCORESET approach was not developed for efficient learning but instead was developed to capture coresets that try to represent the log-likelihood of the entire dataset that MCMC methods can further use. We would also like to point out that we used the original code implementation of BAYESIANCORESET that is not meant for GPU usage in our

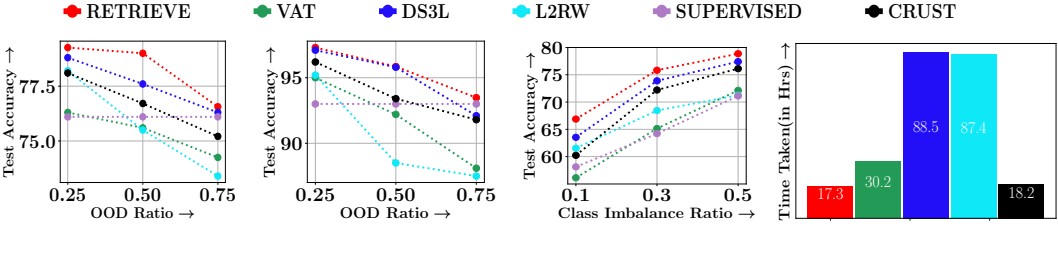

Figure 7: Subfigures (a) to (c) compare the performance of RETRIEVE with baselines (including DS3L and L2RW) for different OOD ratios (a,b) and imbalance ratios (c). We see that RETRIEVE outperforms both baselines in most of the cases. Furthermore, RETRIEVE achieves this while being close to $2\times$ faster than VAT (the SSL algorithm) and $5\times$ faster than the robust SSL algorithms (DS3L and L2RW).

experiments. Hence the speedups of the BAYESIANCORESET approach can be further improved with efficient code implementation. Subfigure 4h shows that RETRIEVE achieves faster convergence compared to all other methods on CIFAR10 for 30% subset with Mean-Teacher. Subfigure 4g shows the extended convergence of RETRIEVE on CIFAR10 for 20% and 30% subsets using VAT, where the RETRIEVE is allowed to train for larger epochs to achieve comparable accuracy with Full training at the cost of losing some efficiency. Note that the points marked by * in subfigure 4g denote the actual training endpoint, i.e. the usual number of epochs/iterations used to obtain points in subfigures (4a, 4b, 4d, 4e, 4c). We observe that RETRIEVE matches the performance of VAT while being close to $2\times$ faster in running times (and correspondingly energy efficiency). We repeat this experiment with MT in the Appendix G. Also, more detailed results with additional convergence plots and tradeoff curves are in Appendix G.

**Robust SSL Results:** We test the performance of RETRIEVE on CIFAR10 and MNIST datasets with OOD in the unlabeled set and CIFAR10 dataset with the class imbalance in both labeled and unlabeled sets. sub-figures 7a, 7b shows the accuracy plots of RETRIEVE for different OOD ratios of 25%, 50% and 75%. The results show that RETRIEVE with VAT outperforms all other baselines, including DS3L [17], a state-of-the-art robust SSL baseline in the OOD scenario. Next, sub-figure 7c shows the accuracy plots of RETRIEVE for different class imbalance ratios of 10%, 30% and 50% on CIFAR-10 dataset. The results show that RETRIEVE with VAT outperforms all other baselines, including DS3L [17] (also run with VAT) in the class imbalance scenario as well. In particular, RETRIEVE outperforms other baselines by at least 1.5% on the CIFAR-10 with imbalance. Sub-figure 7d shows the time taken by different algorithms on the CIFAR10 dataset with a 50% class imbalance ratio. The results show that CRUST did not perform well in terms of accuracy and speedups achieved compared to RETRIEVE. Except for MixUP, CRUST is similar to CRAIG, which did not perform well compared to RETRIEVE in a traditional SSL setting. Furthermore, the performance gain due to MixUP for coreset selection in the SSL setting is minimal. The minimal gain can be attributed to the fact that the hypothesized labels used for MixUP in the earlier stages of training are noisy. Furthermore, as stated earlier, CRUST was developed to tackle noisy labels in a supervised learning setting and is not developed to deal with OOD or Class Imbalance in general. The results show that RETRIEVE is more efficient compared to the other baselines. In particular, RETRIEVE is 5x times faster compared to DS3L method. Other detailed results (tradeoff curves and convergence curves) are in Appendix H.

## 5 Conclusion and Broader Impacts

We introduce RETRIEVE, a discrete-continuous bi-level optimization based coreset selection method for efficient and robust semi-supervised learning. We show connections with weak-submodularity, which enables the coreset selection in RETRIEVE to be solved using a scalable stochastic greedy algorithm. Empirically, we show that RETRIEVE is very effective for SSL. In particular, it achieves $3\times$ speedup on a range of SSL approaches like VAT, MT, and FixMatch with around $0.7\%$ accuracy loss and a $2\times$ speedup with no accuracy loss. In the case of robust SSL with imbalance and OOD data, RETRIEVE outperforms existing SOTA methods while being $5\times$ faster. We believe RETRIEVE has a significant positive societal impact by making SSL algorithms (specifically robust SSL) significantly faster and more energy-efficient, thereby reducing the $CO_2$ emissions incurred during training.

# 6 Acknowledgments and Disclosure of Funding

We would like to thank NeurIPS area chairs and anonymous reviewers for their efforts in reviewing this paper and their constructive comments! RI and KK were funded by the National Science Foundation(NSF) under Grant Number 2106937, a startup grant from UT Dallas, and a Google and Adobe research award. FC and XZ were funded by National Science Foundation(NSF) under Grant Numbers 1815696, 1750911, and 2107449. Any opinions, findings, and conclusions or recommendations expressed in this material are those of the author(s) and do not necessarily reflect the views of the National Science Foundation, Google or Adobe.

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
