# Supplementary Material

# Appendix

## Table of Contents

## A Code and Licenses

### A.1 Code

The code of RETRIEVE for VAT, MT is available at the following link: `https://github.com/decile-team/cords`. The code of RETRIEVE for FixMatch is available at the following link: `https://github.com/krishnatejakk/EfficientFixMatch`. We will transfer the FixMatch code to the CORDS repository to have a unified repository shortly.

### A.2 Licenses

We release both the code repositories of RETRIEVE with MIT license, and it is available for everybody to use freely. For MT and VAT, we built upon the open-source Pytorch implementation[4] which is an MIT licensed repo. For the FixMatch method, we implemented it based on an open-source Pytorch implementation[5]. For DS3L [17], we implemented it based on the released code [6] which has an unknown license. Nevertheless, the authors of the DS3L [17] made the code available for everyone to use. For L2RW [49], we used the open-source Pytorch implementation[7] which has an unknown license and adapted it to the SSL settings. Nevertheless, the owner of the repository made the code available for everyone to use.

As far as the datasets are considered, we use CIFAR10 [27], SVHN [43] and MNIST [32] datasets. CIFAR10 dataset is released with an MIT license. MNIST dataset is released with an Creative Commons Attribution-Share Alike 3.0 license. SVHN dataset is released with a CC0:Public Domain license. All the datasets used in this work are publicly available. Furthermore, the datasets used do not contain any personally identifiable information.

## B Proof of Theorem 1

We begin by first stating and then proving Theorem 1.

**Theorem** *Optimization problem (Equation (5)) is NP hard, even if $l_s$ is a convex loss function. If the labeled set loss function $l_s$ is cross-entropy loss, then the optimization problem give in the Equation (5) can be converted into an instance of cardinality constrained weakly submodular maximization.*

We use the proof techniques similar to the ones used in Theorem-1 of GLISTER [24]. GLISTER proved the weak-submodularity only for the case when both the loss functions in the bi-level optimization problem are cross-entropy losses. In our work, we prove the weak submodularity with the SSL objective as an inner level loss. Furthermore, we prove the weak-submodularity when the unlabeled set loss is either cross-entropy loss or squared loss functions.

### B.1 Proof Sketch

We introduce the notations used in the proof of the theorem in subsection B.2. In our proof, we prove that the optimization problem given in Equation (5) is an $\alpha$-submodular function. We give the definitions of $\alpha$-submodularity, and the approximation guarantees achieved by greedy algorithms in subsection B.3. We state the lemmas of the $\alpha$-submodularity satisfied by the RETRIEVE framework for different cases in subsection B.4. Finally, we give the proof of $\alpha$-submodularity of the RETRIEVE when the unlabeled set loss is cross-entropy loss or squared loss in the subsection B.5.

### B.2 Notation

Let $(x_i, y_i)$ be the $i^{th}$ data-point in the labeled set $\mathcal{D}$ where $i \in [1, n]$, and $(x_i^t)$ be the $i^{th}$ data-point in the unlabeled set $\mathcal{U}$ where $i \in [1, m]$. Consider a classification task with $C$ classes. Let the classifier model be characterized by the model parameters $\theta$. As shown in the Equation (6), the set function

---

[4] `https://github.com/perrying/pytorch-consistency-regularization`
[5] `https://github.com/kekmodel/FixMatch-pytorch`
[6] `https://github.com/guolz-ml/DS3L`
[7] `https://github.com/danieltan07/learning-to-reweight-examples`

of RETRIEVE is given by $f(\theta_t, \mathcal{S}) = -L_S(\mathcal{D}, \theta_t - \alpha_t \nabla_\theta L_S(\mathcal{D}, \theta_t) - \alpha_t \lambda_t \sum_{j \in \mathcal{S}} \boldsymbol{m}_{jt} \nabla_\theta l_u(x_j, \theta_t))$

where $L_S$ is the cross-entropy loss.

The coreset selection optimization problem of RETRIEVE can be written as follows:

$$\mathcal{S}_{t+1} = \underset{\mathcal{S} \subseteq \mathcal{U}, |\mathcal{S}| = k}{\operatorname{argmax}} f(\theta_t, \mathcal{S}) \tag{8}$$

Denote the gain of a set function as $G_{\theta, v}(s) = f(\theta, v \cup s) - f(\theta, v)$. In this proof, we prove that the above optimization problem is approximately submodular [12].

### B.3   $\alpha$-submodularity

Here, we discuss some prior works on submodularity. The definition of $\alpha$-submodularity is given below:

**Definition:** A function is called $\alpha$-submodular [15], if the gain of adding an element $e$ to set $X$ is $1 - \alpha$ times greater than or equals to the gain of adding an element $e$ to set $Y$ where $X \subseteq Y$. i.e.,

$$\underset{X, Y | X \subseteq Y}{\forall} G_{\theta, X}(e) \geq (1 - \alpha) G_{\theta, Y}(e) \tag{9}$$

This definition is different from the notion of $\gamma$-weakly submodular functions [12]. However, as stated in the Proposition 4 of [15], $\alpha$-submodular functions and $\gamma$-weakly submodular functions are closely related, where the function that is $\alpha$-submodular is also $\gamma$-weakly submodular with the submodularity ratio $\gamma \geq 1 - \alpha$.

This further implies the following approximation guarantee given below:

**Lemma 1** *[12, 15] Given a $\alpha$-approximate monotone submodular function $f$, the greedy algorithm achieves a $1 - e^{-(1-\alpha)}$ approximation factor for the problem of maximizing $f$ subject to cardinality constraints.*

Next, we show that $f(\theta_t, \mathcal{S})$ is a $\alpha$-approximate submodular function. To this extent, we assume that the norm of data points in the labeled and unlabeled sets are bounded such that $\|x_i\| \leq R$. Note that this is common assumption made in most convergence analysis results. When $l_s$ is cross entropy loss function, we prove that the set function $f(\theta_t, \mathcal{S})$ of RETRIEVE is $\alpha$-approximate submodular function where $\alpha = \frac{2R^2}{2R^2 + 1}$.

### B.4   $\alpha$-submodularity of RETRIEVE

We now show the $\alpha$ submodularity of the set function for different cases of the unlabeled loss $l_u$.

**Lemma 2** *If the labeled set loss function $l_s$ is the cross entropy loss and the unlabeled set loss function $l_u$ is the cross entropy loss, then the optimization problem given in equation Equation (5) is an instance of cardinality constrained $\alpha$-approximate submodular maximization, where $\alpha = \frac{2R^2}{2R^2 + 1}$ such that $R$ is the maximum l-2 norm of the data instances in both labeled and the unlabeled sets.*

**Lemma 3** *If the labeled set loss function $l_s$ is the cross entropy loss and the unlabeled set loss function $l_u$ is the squared loss, then the optimization problem given in equation Equation (5) is an instance of cardinality constrained $\alpha$-approximate submodular maximization, where $\alpha = \frac{R^2}{R^2 + 1}$ such that $R$ is the maximum l-2 norm of the data instances in both labeled and the unlabeled sets.*

### B.5   Proof

Assuming that we start at $\theta_0$ and for the ease of notation we use $f(\mathcal{S})$ instead of $f(\theta_0, \mathcal{S})$. When $l_s$ is cross entropy loss, the optimization problem given in Equation (5) can be written as follows:

$$\begin{aligned} \mathcal{S}_1 &= \underset{\mathcal{S} \subseteq \mathcal{U}: |\mathcal{S}| \leq k}{\operatorname{argmax}} f(\mathcal{S}) \\ &= \underset{\mathcal{S} \subseteq \mathcal{U}: |\mathcal{S}| \leq k}{\operatorname{argmax}} -L_S(\mathcal{D}, \theta_0 - \alpha_0 \nabla_\theta L_S(\mathcal{D}, \theta_0) - \alpha_0 \lambda_0 \sum_{j \in \mathcal{S}} \boldsymbol{m}_{j0} \nabla_\theta l_u(x_j, \theta_0)) \end{aligned} \tag{10}$$

Substituting $l_s$ with the cross-entropy loss function, then the set function $f(\mathcal{S})$ can be written as follows:

$$f(\mathcal{S}) = \sum_{i=1}^{n} \log \left( \frac{\exp((\theta_0^{y_i} - \alpha_0 \nabla_\theta L_S(\mathcal{D}, \theta_0^{y_i}) - \alpha_0 \lambda_0 \sum_{j \in \mathcal{S}} \boldsymbol{m}_{j0} \nabla_\theta l_u(x_j, \theta_0^{y_i}))^T x_i)}{\sum_{c \in [1,C]} \exp((\theta_0^c - \alpha_0 \nabla_\theta L_S(\mathcal{D}, \theta_0^{y_i}) - \alpha_0 \lambda_0 \sum_{j \in \mathcal{S}} \boldsymbol{m}_{j0} \nabla_\theta l_u(x_j, \theta_0^c))^T x_i)} \right) \quad (11)$$

Rewriting the above equation, we achieve:

$$f(\mathcal{S}) = \sum_{i=1}^{n} \left( \log \left( \exp((\theta_0^{y_i} - \alpha_0 \nabla_\theta L_S(\mathcal{D}, \theta_0^{y_i}) - \alpha_0 \lambda_0 \sum_{j \in \mathcal{S}} \boldsymbol{m}_{j0} \nabla_\theta l_u(x_j, \theta_0^{y_i}))^T x_i) \right) \right.$$
$$\left. - \log \left( \sum_{c \in [1,C]} \exp((\theta_0^c - \alpha_0 \nabla_\theta L_S(\mathcal{D}, \theta_0^{y_i}) - \alpha_0 \lambda_0 \sum_{j \in \mathcal{S}} \boldsymbol{m}_{j0} \nabla_\theta l_u(x_j, \theta_0^c))^T x_i) \right) \right) \quad (12)$$

$$f(\mathcal{S}) = \sum_{i=1}^{n} \left( (\theta_0^{y_i} - \alpha_0 \nabla_\theta L_S(\mathcal{D}, \theta_0^{y_i}) - \alpha_0 \lambda_0 \sum_{j \in \mathcal{S}} \boldsymbol{m}_{j0} \nabla_\theta l_u(x_j, \theta_0^{y_i}))^T x_i \right. \quad (13)$$
$$\left. - \log \left( \sum_{c \in [1,C]} \exp((\theta_0^c - \alpha_0 \nabla_\theta L_S(\mathcal{D}, \theta_0^{y_i}) - \alpha_0 \lambda_0 \sum_{j \in \mathcal{S}} \boldsymbol{m}_{j0} \nabla_\theta l_u(x_j, \theta_0^c))^T x_i) \right) \right) \quad (14)$$

Since, the term $(\theta_0^{y_i} - \alpha_0 \nabla_\theta L_S(\mathcal{D}, \theta_0^{y_i}))^T x_i$ does not depend on the subset $S$, we can remove it from our optimization problem,

$$f(\mathcal{S}) = \sum_{i=1}^{n} \left( \sum_{j \in \mathcal{S}} -\alpha_0 \lambda_0 \boldsymbol{m}_{j0} \nabla_\theta l_u(x_j, \theta_0^{y_i})^T x_i \right. \quad (15)$$
$$\left. - \log \left( \sum_{c \in [1,C]} \exp((\theta_0^c - \alpha_0 \nabla_\theta L_S(\mathcal{D}, \theta_0^{y_i}) - \alpha_0 \lambda_0 \sum_{j \in \mathcal{S}} \boldsymbol{m}_{j0} \nabla_\theta l_u(x_j, \theta_0^c))^T x_i) \right) \right) \quad (16)$$

Assume $g_{ijc} = \boldsymbol{m}_{j0} \nabla_\theta l_u(x_j, \theta_0^c))^T x_i$,

$$f(\mathcal{S}) = \sum_{i=1}^{n} \left( \sum_{j \in \mathcal{S}} -\alpha_0 \lambda_0 g_{ijy_i} - \log \left( \sum_{c \in [1,C]} \exp((\theta_0^c - \alpha_0 \nabla_\theta L_S(\mathcal{D}, \theta_0^{y_i}))^T x_i - \alpha_0 \lambda_0 \sum_{j \in \mathcal{S}} g_{ijc}) \right) \right) \quad (17)$$

$$f(\mathcal{S}) = \sum_{i=1}^{n} \left( \sum_{j \in \mathcal{S}} -\alpha_0 \lambda_0 g_{ijy_i} - \log \left( \sum_{c \in [1,C]} \exp((\theta_0^c - \alpha_0 \nabla_\theta L_S(\mathcal{D}, \theta_0^{y_i}))^T x_i) \exp(-\alpha_0 \lambda_0 \sum_{j \in \mathcal{S}} g_{ijc}) \right) \right) \quad (18)$$

Let $h_{ic} = \exp((\theta_0^c - \alpha_0 \nabla_\theta L_S(\mathcal{D}, \theta_0^{y_i}))^T x_i)$ where $h_{ic} \geq 0$ as $h_{ic}$ is an exponential function.

$$f(\mathcal{S}) = \sum_{i=1}^{n} \left( \sum_{j \in \mathcal{S}} -\alpha_0 \lambda_0 g_{ijy_i} - \log \left( \sum_{c \in [1,C]} h_{ic} \exp(-\alpha_0 \lambda_0 \sum_{j \in \mathcal{S}} g_{ijc}) \right) \right) \quad (19)$$

As $g_{ijc}$ is not always greater than zero, we can make some transformations to convert the problem into a monotone submodular function. First, we transform $g_{ijc}$ to $\hat{g}_{ijc}$ such that $g_{ijc} = \hat{g}_{ijc} + g_m - 1$ where $g_m = \min_{i,j,c} g_{ijc}$. This transformation ensures that $\hat{g}_{ijc} \geq 1$. Denote $g_{nm} = \min_{i,j,c}(-g_{ijc})$, and then we define a transformation of $g_{ijc}$ to $g_{ijc}''$ such that $-g_{ijc} = g_{ijc}'' + g_{nm}$. Note that both $\hat{g}_{ijc}$ and $g_{ijc}''$ are greater than or equal to zero after the transformations.

$$f(\mathcal{S}) = \sum_{i=1}^{n} \left( \sum_{j \in \mathcal{S}} \alpha_0 \lambda_0 (g''_{ijy_i} + g_{nm}) - \log \left( \sum_{c \in [1,C]} h_{ic} \exp(-\alpha_0 \lambda_0 \sum_{j \in \mathcal{S}} (\hat{g}_{ijc} + g_m - 1)) \right) \right)$$

(20)

$$= \alpha_0 \lambda_0 k n g_{nm} + \sum_{i=1}^{n} \left( \sum_{j \in \mathcal{S}} \alpha_0 \lambda_0 g''_{ijy_i} - \log \left( \sum_{c \in [1,C]} h_{ic} \exp(-\alpha_0 \lambda_0 \sum_{j \in \mathcal{S}} \hat{g}_{ijc}) \exp(k(g_m - 1)) \right) \right)$$

(21)

where $k$ is the size of the subset.

Denote $H_{ic} = h_{ic} \exp(k(g_m - 1))$. Further as $\alpha_0 \lambda_0 n k(g_{nm})$ is a constant, we can remove it from the optimization problem and we can define the new optimization set function $\hat{f}(\mathcal{S})$ as shown below:

$$\hat{f}(\mathcal{S}) = \sum_{i=1}^{n} \sum_{j \in \mathcal{S}} \alpha_0 \lambda_0 (g''_{ijy_i}) - \sum_{i=1}^{n} \log \left( \sum_{c \in [1,C]} H_{ic} \exp(-\alpha_0 \lambda_0 \sum_{j \in \mathcal{S}} \hat{g}_{ijc}) \right)$$

(22)

In the above equation, denote the first part as $f_1(\mathcal{S}) = \sum_{i=1}^{n} \sum_{j \in S} \alpha_0 \lambda_0 (g''_{ijy_i})$ which is a monotone modular function in $\mathcal{S}$. Similarly, denote the second part $f_2(\mathcal{S}) = -\sum_{i=1}^{n} \log \left( \sum_{c \in [1,C]} H_{ic} \exp(-\alpha_0 \lambda_0 \sum_{j \in \mathcal{S}} (\hat{g}_{ijc})) \right)$ is a monotone function but is not submodular.

Hence, we prove that the function $f_2(\mathcal{S})$ is an $\alpha$-submodular function in the following proof section. Furthermore, since the first part is positive modular, it is easy to see that if $f_2(X)$ is $\alpha$ submodular (with $\alpha \geq 0$), then the function $f_1(X) + f_2(X)$ will also be an $\alpha$-submodular function.

Note that, a function $h(X)$ is $\alpha$-submodular if $h(j|X) \geq (1 - \alpha)h(j|Y)$ for all subsets $X \subseteq Y$. Assuming that $f_2$ is $\alpha$-submodular, then the following holds:

$$f_2(j|X) \geq (1 - \alpha)f_2(j|Y)$$

for all subsets $X \subseteq Y$.

As $f_1$ is positive modular, we have the following:

$$f_1(j|X) = f_1(j|Y) \geq (1 - \alpha)f_1(j|Y)$$

.

This implies that the function $f_1(\mathcal{S}) + f_2(\mathcal{S})$ is an $\alpha$-submodular function which further implies that it is also an $\gamma$-weakly submodular function.

$\alpha$**-submodularity proof of function** $f_2$**:**

The gain of adding an element $e$ to the set $X$ is given as follows:

$$f_2(e|X) = -\sum_{i=1}^{n} \log \left( \sum_{c \in [1,C]} H_{ic} \exp(-\alpha_0 \lambda_0 \sum_{j \in \mathcal{S} \cup e} \hat{g}_{ijc}) \right)$$
$$+ \sum_{i=1}^{n} \log \left( \sum_{c \in [1,C]} H_{ic} \exp(-\alpha_0 \lambda_0 \sum_{j \in \mathcal{S}} \hat{g}_{ijc}) \right)$$

(23)

$$f_2(e|X) = -\sum_{i=1}^{n} \log \left( \sum_{c \in [1,C]} H_{ic} \exp(-\alpha_0 \lambda_0 \sum_{j \in \mathcal{S}} \hat{g}_{ijc} - \alpha_0 \lambda_0 \hat{g}_{iec}) \right)$$
$$+ \sum_{i=1}^{n} \log \left( \sum_{c \in [1,C]} H_{ic} \exp(-\alpha_0 \lambda_0 \sum_{j \in \mathcal{S}} \hat{g}_{ijc}) \right)$$

(24)

Let, $\hat{g}_m = \min_{ijc} \hat{g}_{ijc}$. Then, we can rewrite the above equation as following:

$$f_2(e|X) \geq -\sum_{i=1}^{n} \log\left(\sum_{c\in[1,C]} H_{ic}\exp(-\alpha_0\lambda_0\sum_{j\in\mathcal{S}}\hat{g}_{ijc} - \alpha_0\lambda_0\hat{g}_m)\right)$$
$$+ \sum_{i=1}^{n} \log\left(\sum_{c\in[1,C]} H_{ic}\exp(-\alpha_0\lambda_0\sum_{j\in\mathcal{S}}\hat{g}_{ijc})\right) \tag{25}$$

$$f_2(e|X) \geq -\sum_{i=1}^{n} \log\left(\sum_{c\in[1,C]} H_{ic}\exp(-\alpha_0\lambda_0\sum_{j\in\mathcal{S}}\hat{g}_{ijc})\exp(-\alpha_0\lambda_0\hat{g}_m)\right) \tag{26}$$

$$+ \sum_{i=1}^{n} \log\left(\sum_{c\in[1,C]} H_{ic}\exp(-\alpha_0\lambda_0\sum_{j\in\mathcal{S}}\hat{g}_{ijc})\right) \tag{27}$$

$$f_2(e|X) \geq -\sum_{i=1}^{n} \log\left(\sum_{c\in[1,C]} H_{ic}\exp(-\alpha_0\lambda_0\sum_{j\in\mathcal{S}}\hat{g}_{ijc})\right) + \sum_{i=1}^{n} \alpha_0\lambda_0\hat{g}_m \tag{28}$$

$$+ \sum_{i=1}^{n} \log\left(\sum_{c\in[1,C]} H_{ic}\exp(-\alpha_0\lambda_0\sum_{j\in\mathcal{S}}\hat{g}_{ijc})\right) \tag{29}$$

$$f_2(e|X) \geq n\alpha_0\lambda_0\hat{g}_m \tag{30}$$

Let, $\hat{g}_{max} = \max_{ijc} \hat{g}_{ijc}$. Then, we can rewrite the Equation (24) as following:

$$f_2(e|X) \leq -\sum_{i=1}^{n} \log\left(\sum_{c\in[1,C]} H_{ic}\exp(-\alpha_0\lambda_0\sum_{j\in\mathcal{S}}\hat{g}_{ijc} - \alpha_0\lambda_0\hat{g}_{max})\right)$$
$$+ \sum_{i=1}^{n} \log\left(\sum_{c\in[1,C]} H_{ic}\exp(-\alpha_0\lambda_0\sum_{j\in\mathcal{S}}\hat{g}_{ijc})\right) \tag{31}$$

$$f_2(e|X) \leq -\sum_{i=1}^{n} \log\left(\sum_{c\in[1,C]} H_{ic}\exp(-\alpha_0\lambda_0\sum_{j\in\mathcal{S}}\hat{g}_{ijc})\exp(-\alpha_0\lambda_0\hat{g}_{max})\right) \tag{32}$$

$$+ \sum_{i=1}^{n} \log\left(\sum_{c\in[1,C]} H_{ic}\exp(-\alpha_0\lambda_0\sum_{j\in\mathcal{S}}\hat{g}_{ijc})\right) \tag{33}$$

$$f_2(e|X) \leq -\sum_{i=1}^{n} \log\left(\sum_{c\in[1,C]} H_{ic}\exp(-\alpha_0\lambda_0\sum_{j\in\mathcal{S}}\hat{g}_{ijc})\right) + \sum_{i=1}^{n} \alpha_0\lambda_0\hat{g}_{max} \tag{34}$$

$$+ \sum_{i=1}^{n} \log\left(\sum_{c\in[1,C]} H_{ic}\exp(-\alpha_0\lambda_0\sum_{j\in\mathcal{S}}\hat{g}_{ijc})\right) \tag{35}$$

$$f_2(e|X) \leq n\alpha_0\lambda_0\hat{g}_{max} \tag{36}$$

Using the minimum bounds and the maximum bounds given in Equation (30) and Equation (36) on $f_2(e|X)$, we have:

$$\underset{X,Y|X\subseteq Y}{\forall} \frac{f_2(e|X)}{f_2(e|Y)} \geq \frac{\hat{g}_m}{\hat{g}_{max}} \tag{37}$$

Since, $g_{ijc} = \boldsymbol{m}_{j0} \nabla_\theta l_u(x_j, \theta_0^c)^T x_i$, and $\hat{g}_{ijc} = g_{ijc} - g_m + 1$, we have:

$$\hat{g}_{ijc} = \boldsymbol{m}_{j0} \nabla_\theta l_u(x_j, \theta_0^c)^T x_i - \min_{i,j,c} \boldsymbol{m}_{j0} \nabla_\theta l_u(x_j, \theta_0^c))^T x_i + 1 \tag{38}$$

For most consistency based SSL algorithms, $l_u$ is either cross-entropy loss or mean-squared loss on the hypothesized label probability prediction.

$\alpha$-submodularity when $l_u$ is a cross-entropy loss function: Let $[p_{j1}, \cdots, p_{jC}]$ be the class probabilities output by the model for instance $x_j$ in the unlabeled set after the softmax operator and $[q_{j1}, \cdots, q_{jC}]$ be the target probability. If $l_u$ is a cross-entropy loss function, we know that $\nabla_\theta l_u(x_j, \theta_0^c) = \sum_{k=1}^C q_{jk}(1_{k=c} - p_{jk})x_j$ where $1_{k=c} = 1$ if $k = c$ and $0$ otherwise. Hence,

$$\mathop{\forall}_{i,j,c} \nabla_\theta \boldsymbol{m}_{j0} l_u((x_j^t, \theta_0^c))^T x_i \leq R^2 \quad \text{where} \quad R \geq \mathop{\forall}_i \left\| x_i^t \right\| \tag{39}$$

Similarly,

$$\mathop{\forall}_{i,j,c} \nabla_\theta \boldsymbol{m}_{j0} l_u((x_j^t, \theta_0))^T x_i \geq -R^2 \quad \text{where} \quad R \geq \mathop{\forall}_i \left\| x_i^t \right\| \tag{40}$$

Similarly, norm of the labeled set points are bounded from above by $R$. Therefore. $\hat{g}_m = 1$ and $\hat{g}_{max} = 2R^2 + 1$. This implies that:

$$\mathop{\forall}_{X,Y|X \subseteq Y} \frac{f_2(e|X)}{f_2(e|Y)} \geq \frac{1}{2R^2 + 1} \tag{41}$$

Since $1 - \alpha = \frac{1}{2R^2+1}$, then $\alpha = \frac{2R^2}{2R^2+1}$.

$\alpha$-submodularity when $l_u$ is a squared loss function: Let $[p_{j1}, \cdots, p_{jC}]$ be the class probabilities output by the model for instance $x_j$ in the unlabeled set after the softmax operator and $[q_{j1}, \cdots, q_{jC}]$ be the target probability. If $l_u$ is a squared loss function, we know that $\nabla_{\theta_0^c} l_u(x_j, \theta_0) = \sum_{k=1}^C 2(q_{jk} - p_{jk})(1_{k=c} - p_{jk})p_{jk}x_j$ where $1_{k=c} = 1$ if $k = c$ and $0$ otherwise.

$$\mathop{\forall}_{i,j,c} \nabla_\theta \boldsymbol{m}_{j0} l_u((x_j^t, \theta_0))^T x_i \leq R^2/2 \quad \text{where} \quad R \geq \mathop{\forall}_i \left\| x_i^t \right\| \tag{42}$$

Similarly,

$$\mathop{\forall}_{i,j,c} \nabla_\theta \boldsymbol{m}_{j0} l_u((x_j^t, \theta_0))^T x_i \geq -R^2/2 \quad \text{where} \quad R \geq \mathop{\forall}_i \left\| x_i^t \right\| \tag{43}$$

Similarly, norm of the labeled set points are bounded from above by $R$. Therefore. $\hat{g}_m = 1$ and $\hat{g}_{max} = R^2 + 1$. This implies that:

$$\mathop{\forall}_{X,Y|X \subseteq Y} \frac{f_2(e|X)}{f_2(e|Y)} \geq \frac{1}{R^2 + 1} \tag{44}$$

Since $1 - \alpha = \frac{1}{2R^2+1}$, then $\alpha = \frac{R^2}{R^2+1}$.

From both the cases, this implies that $f_2$ is $\alpha$ submodular having $\alpha = \frac{2R^2}{2R^2+1}$ when $l_u$ is cross-entropy loss and $\alpha = \frac{R^2}{R^2+1}$ when $l_u$ is squared loss, which further implies that $f$ is $\alpha$-submodular. This further implies that, any greedy algorithm will achieve a $1 - e^{-(1-\alpha)}$ approximation factor, for the coreset selection step when $l_s$ is a cross-entropy loss and $l_u$ is squared loss or cross-entropy loss.

Finally, the proof of the NP-hardness of the mixed discrete-continuous bi-level optimization problem is shown in Lemma-1 of the work [24].

# C  Loss formulations for different SSL algorithms

## C.1  Notation

Denote $\mathcal{D} = \{x_i, y_i\}_{i=1}^n$ to be the labeled set with $n$ labeled data points, and $\mathcal{U} = \{x_j\}_{j=1}^m$ to be the unlabeled set with $m$ data points. Let $\theta$ be the classifier model parameters, $l_s$ be the labeled set loss function (such as cross-entropy loss) and $l_u$ be the unlabeled set loss, e.g. consistency-regularization loss, entropy loss, etc.. Denote $L_S(\mathcal{D}, \theta) = \sum\limits_{i \in \mathcal{D}} l_s(\theta, x_i, y_i)$ and $L_U(\mathcal{U}, \theta, \boldsymbol{m}) = \sum\limits_{j \in \mathcal{U}} \boldsymbol{m}_j l_u(x_j, \theta)$ where $\boldsymbol{m} \in \{0, 1\}^m$ is the binary mask vector for unlabeled set. For notational convenience, we denote $l_{si}(\theta) = l_s(x_i, y_i, \theta)$ and denote $l_{uj}(\theta) = l_u(x_j, \theta)$. We also assume that the functions $l_s$ and $l_u$ involves the scaling constants like $\frac{1}{n}, \frac{1}{m}$ required to consider other loss reductions like mean loss.

**Semi-supervised loss:**  Following the above notations, the loss function for many existing SSL algorithms can be written as $L_S(\mathcal{D}, \theta) + \lambda L_U(\mathcal{U}, \theta, \boldsymbol{m})$, where $\lambda$ is the regularization coefficient for the unlabeled set loss. For Mean Teacher [56], VAT [42], MixMatch [4], the mask vector $\boldsymbol{m}$ is made up entirely of ones, whereas for FixMatch [53], $\boldsymbol{m}$ is confidence-thresholded binary vector, indicating whether to include an unlabeled data instance or not. Usually, $L_S$ is a cross-entropy loss for classification experiments and squared loss for regression experiments.

Detailed description of SSL loss formulation for different SSL algorithms are given below:

## C.2  Mean-Teacher

Mean Teacher [56] proposed to generate a more stable target output for data points in the unlabeled set using the output of the model using the exponential moving average of model parameter values at previous iterations. Denote the exponential moving average of model parameters as $\hat{\theta}$. Further, denote $f(\theta, x_i)$ as the softmax of the logits of the datapoint $x_i$ obtained from the model with model parameters $\theta$.

The loss function of Mean-Teacher algorithm is as follows:

$$L_S(\mathcal{D}, \theta) + \lambda \sum_{j=1}^m \frac{1}{m} \| f(\theta, x_j) - f(\hat{\theta}, x_j) \|_2^2$$

where $L_S$ is the mean cross-entropy loss for classification experiments. Further, the mask vector $\boldsymbol{m}$ in the case of Mean-Teacher algorithm is made up entirely of ones. And the unlabeled set loss function is a squared loss function.

## C.3  VAT

Virtual adversarial training(VAT) [42] tries to find the additional perturbation to the unlabeled data points such that the KL divergence loss is maximized with respect to class predictions distribution after the perturbation.

Let, $f$ be the classifier model characterized by the model parameters $\theta$. Let, $d$ be the additive perturbation to the unlabeled set. Let, $KL(p, q)$ be the KL-Divergence loss between distributions $p$ and $q$. Further, denote $f(\theta, x_i)$ as the softmax of the logits of the datapoint $x_i$ obtained from the model with model parameters $\theta$.

Then, the additional perturbation is given as follows:

$$\hat{d} = \operatorname*{argmax}_d \sum_{j=1}^m \frac{1}{m} KL(f(\theta, x_j), f(\theta, x_j + d))$$

The loss function of VAT algorithm is as follows:

$$L_S(\mathcal{D}, \theta) + \lambda \sum_{j=1}^m \frac{1}{m} KL(f(\theta, x_j), f(\theta, x_j + \hat{d}))$$

where $L_S$ is the mean cross-entropy loss for classification experiments. Further, the mask vector $\boldsymbol{m}$ in the case of VAT algorithm is made up entirely of ones. And the unlabeled set loss function is a KL divergence loss function.

## C.4 MixMatch

MixMatch [4] performs augmentations on unlabeled instances and gets a pseudo-label prediction after sharpening the average predictions with different augmentations like shifts, cropping, image flipping, weak and strong augmentation to design the regularization function. Finally, the augmented labeled set and unlabeled sets are concatenated and shuffled to form a new dataset which is used in mix-up [61].

Let, $f$ be the classifier model characterized by the model parameters $\theta$. Let, $(\hat{x}_i, \hat{p}_i)_{i \in [1,n]}$ be the labeled set after mix-up and $(\hat{x}_j, \hat{p}_j)_{j \in [1,m]}$ be the unlabeled set after mix-up with predicted labels. Further, denote $f(\theta, x_i)$ as the softmax of the logits of the datapoint $x_i$ obtained from the model with model parameters $\theta$.

The loss function of Mix-Match algorithm is as follows:

$$\frac{1}{n}\sum_{i=1}^{n} CE(f(\theta, \hat{x}_i), \hat{p}_i) + \lambda \sum_{j=1}^{m} \frac{1}{m}\|f(\theta, \hat{x}_j) - \hat{p}_j\|_2^2$$

where $CE(p, q)$ is the cross-entropy loss between distributions $p$ and $q$. Further, the mask vector $\boldsymbol{m}$ in the case of MixMatch algorithm is made up entirely of ones. And the unlabeled set loss function is a l2 squared loss function.

## C.5 FixMatch

FixMatch [53] uses the cross-entropy loss between class predictions of weak augmented and strong augmented data points as the regularization function. Further, FixMatch uses confidence-based thresholding to consider only unlabeled instances with confident model predictions.

Let, $f$ be the classifier model characterized by the model parameters $\theta$. Let, $\hat{x}_i$ be the weakly augmented version of data point $x_i$ and $\hat{x}_i^s$ be the strong augmented version of data point $x_i$. Further, denote $f(\theta, x_i)$ as the softmax of the logits of the datapoint $x_i$ obtained from the model with model parameters $\theta$.

Then the loss function of FixMatch algorithm is as follows:

$$\frac{1}{n}\sum_{i=1}^{n} CE(f(\theta, \hat{x}_i), y_i) + \lambda \sum_{j=1}^{m} 1_{\max(f(\theta, \hat{x}_j)) \geq \tau} \frac{1}{m} CE(f(\theta, \hat{x}_j), f(\theta, \hat{x}_j^s))$$

where $CE(p, q)$ is the cross-entropy loss between distributions $p$ and $q$. Further, the mask vector $\boldsymbol{m}$ in the case of FixMatch algorithm is a binary vector based on confidence thresholding i.e., $\boldsymbol{m}_j = 1_{\max(f(\theta, \hat{x}_j))}$. And the unlabeled set loss function is also a cross-entropy loss between the weakly and strongly augmented versions.

## D  CRAIG Algorithm for SSL

In this section, we discuss the formulation of CRAIG [39] for coreset selection in the semi-supervised learning scenario. CRAIG tries to select a coreset of the unlabeled set $\mathcal{U}$ such that the unlabeled loss gradient on the entire unlabeled set is equal to the weighted sum of the unlabeled loss of the individual data points in the selected coreset.

The optimization problem of CRAIG in the semi-supervised learning scenario can be written as follows:

$$\mathcal{S}^* = \operatorname*{argmin}_{\mathcal{S} \subseteq \mathcal{U}:|\mathcal{S}| \leq k, \{\gamma_j\}_{j \in [1,|\mathcal{S}|]}:\forall_j \gamma_j \geq 0} \left\| \sum_{i \in \mathcal{U}} \boldsymbol{m}_i \nabla_\theta l_u(x_i, \theta) - \sum_{j \in \mathcal{S}} \boldsymbol{m}_j \gamma_j \nabla_\theta l_u(x_j, \theta) \right\| \tag{45}$$

Let the objective function of CRAIG be denoted as $f(\mathcal{S}, \theta) = \left\| \sum_{i \in \mathcal{U}} \boldsymbol{m}_i \nabla_\theta l_u(x_i, \theta) - \sum_{j \in \mathcal{S}} \boldsymbol{m}_j \gamma_j \nabla_\theta l_u(x_j, \theta) \right\|$.

The above objective function can be upper bounded by converting it into a k-medoids objective function as shown in CRAIG[39]:

$$f(\mathcal{S}, \theta) = \left\| \sum_{i \in \mathcal{U}} \boldsymbol{m}_i \nabla_\theta l_u(x_i, \theta) - \sum_{j \in \mathcal{S}} \boldsymbol{m}_j \gamma_j \nabla_\theta l_u(x_j, \theta) \right\|$$

$$\leq \sum_{i \in \mathcal{U}} \min_{j \in \mathcal{S}} \left\| \boldsymbol{m}_i \nabla_\theta l_u(x_i, \theta) - \boldsymbol{m}_j \nabla_\theta l_u(x_j, \theta) \right\| \tag{46}$$

Then the coreset selection problem of CRAIG in the semi-supervised learning scenario can be written as follows:

$$\mathcal{S}^* = \operatorname*{argmin}_{\mathcal{S} \subseteq \mathcal{U} : |\mathcal{S}| \leq k} \sum_{i \in \mathcal{U}} \min_{j \in \mathcal{S}} \left\| \boldsymbol{m}_i \nabla_\theta l_u(x_i, \theta) - \boldsymbol{m}_j \nabla_\theta l_u(x_j, \theta) \right\| \tag{47}$$

Then the weights for each data instance in the selected coreset is calculated as follows:

$$\gamma_j = \sum_{i \in \mathcal{U}} 1_{j = \operatorname*{argmin}_{k \in \mathcal{S}} \| \boldsymbol{m}_i \nabla_\theta l_u(x_i, \theta) - \boldsymbol{m}_k \nabla_\theta l_u(x_k, \theta) \|} \tag{48}$$

where $1_x = 1$ if $x = True$ and $1_x = 0$ otherwise.

However, in our experiments, we used a per-batch version of the CRAIG problem discussed above since it is shown to be more effective in work [23]. In the per-batch version, we assume that the unlabeled set is divided into a set of mini-batches denoted by $B^u = \{b_1^u, b_2^u, \cdots, b_{\lfloor m/B \rfloor}^u\}$ where $b_1^u = \{x_i : x_i \in \mathcal{U}\}_{i=1}^B$ is a mini-batch of unlabeled set of size $B$. Further, we select $\lfloor k/B \rfloor$ mini-batches in the per-batch version of CRAIG instead of $k$ data points.

The per-batch version of CRAIG can be given as follows:

$$\mathcal{S}^* = \operatorname*{argmin}_{\mathcal{S} \subseteq B^u : |\mathcal{S}| \leq \lfloor k/B \rfloor} \sum_{i \in B^u} \min_{j \in \mathcal{S}} \left\| \sum_{k \in b_i^u} \boldsymbol{m}_k \nabla_\theta l_u(x_k, \theta) - \sum_{l \in b_j^u} \boldsymbol{m}_l \nabla_\theta l_u(x_l, \theta) \right\| \tag{49}$$

Then the weights for all the data instances in a selected mini-batch $b_j^u$ is calculated as follows:

$$\gamma_j = \sum_{i \in B^u} 1_{j = \operatorname*{argmin}_{k \in \mathcal{S}} \left\| \sum_{p \in b_i^u} \boldsymbol{m}_p \nabla_\theta l_u(x_p, \theta) - \sum_{l \in b_k^u} \boldsymbol{m}_l \nabla_\theta l_u(x_l, \theta) \right\|} \tag{50}$$

As discussed earlier, in our experiments, we use the per-batch versions of CRAIG and the optimization problem is given in Equation (49). Furthermore, the weights for each data instance are calculated as shown in the Equation (50).

## E  GRADMATCH Algorithm for SSL

In this section, we discuss the formulation of GRADMATCH [23] for coreset selection in the semi-supervised learning scenario. GRADMATCH tries to select a coreset of the unlabeled set $\mathcal{U}$ such that the unlabeled loss gradient on the entire unlabeled set is equal to the weighted sum of the unlabeled loss of the individual data points in the selected coreset.

The optimization problem of GRADMATCH in the semi-supervised learning scenario can be written as follows:

$$\mathcal{S}^* = \operatorname*{argmin}_{\mathcal{S} \subseteq \mathcal{U} : |\mathcal{S}| \leq k, \{\gamma_j\}_{j \in [1, |\mathcal{S}|]} : \forall_j \gamma_j \geq 0} \left\| \sum_{i \in \mathcal{U}} \boldsymbol{m}_i \nabla_\theta l_u(x_i, \theta) - \sum_{j \in \mathcal{S}} \boldsymbol{m}_j \gamma_j \nabla_\theta l_u(x_j, \theta) \right\| \tag{51}$$

Let the objective function of GRADMATCH be denoted as $f(\mathcal{S}, \theta) = \left\| \sum_{i \in \mathcal{U}} \boldsymbol{m}_i \nabla_\theta l_u(x_i, \theta) - \sum_{j \in \mathcal{S}} \boldsymbol{m}_j \gamma_j \nabla_\theta l_u(x_j, \theta) \right\|$.

The above objective function can be solved using the Orthogonal Matching Pursuit(OMP) algorithm as shown in GRADMATCH[23].

However, in our experiments, we used a per-batch version of the GRADMATCH problem discussed above since it is shown to be more effective in work [23]. In the per-batch version, we assume that the unlabeled set is divided into a set of mini-batches denoted by $B^u = \{b_1^u, b_2^u, \cdots, b_{\lfloor m/B \rfloor}^u\}$ where $b_1^u = \{x_i : x_i \in \mathcal{U}\}_{i=1}^B$ is a mini-batch of unlabeled set of size $B$. Further, we select $\lfloor k/B \rfloor$ mini-batches in the per-batch version of GRADMATCH instead of $k$ data points.

The per-batch version of GRADMATCH can be given as follows:

$$\mathcal{S}^* = \underset{\mathcal{S} \subseteq B^u : |\mathcal{S}| \leq \lfloor k/B \rfloor}{\operatorname{argmin}} \min \left\| \sum_{i \in B^u} \sum_{k \in b_i^u} \boldsymbol{m}_k \nabla_\theta l_u(x_k, \theta) - \sum_{j \in \mathcal{S}} \sum_{l \in b_j^u} \boldsymbol{m}_l \nabla_\theta l_u(x_l, \theta) \right\| \quad (52)$$

Then the weights and the mini-batches are selected using the Orthogonal Matching Pursuit (OMP) algorithm. As discussed earlier, in our experiments, we use the per-batch versions of GRADMATCH and the optimization problem is given in Equation (52).

# F   More Details on Experimental Setup, Datasets, and Baselines

## F.1   Datasets

### F.1.1   Traditional SSL scenario

| Name | No. of classes | No. samples for training | No. samples for validation | No. samples for testing | No. of features | License |
|------|----------------|--------------------------|----------------------------|-------------------------|-----------------|---------|
| CIFAR10 | 10 | 50,000 | - | 10,000 | 32x32x3 | MIT |
| SVHN | 10 | 73,257 | - | 26,032 | 32x32x3 | CC0:Public Domain |

Table 1: Description of the datasets

| Name | Labeled set size | Unlabeled set size | Test set size | Labeled set batch size | Unlabeled set batch size |
|------|------------------|--------------------|---------------|------------------------|--------------------------|
| CIFAR10 | 4000 | 50,000 | 10,000 | 50 | 50 |
| SVHN | 1000 | 73,257 | 26,032 | 50 | 50 |

Table 2: Dataset Splits used in the traditional SSL scenario

We used various standard datasets, viz., CIFAR10, SVHN, to demonstrate the effectiveness and stability of RETRIEVE in the traditional SSL scenario. The descriptions of the datasets used along with the licenses are given in the Table 1. Furthermore, the labeled, unlabeled, and test data splits for each dataset considered along with the labeled and the unlabeled set batch sizes are given in Table 2. Both CIFAR10 and SVHN datasets are publicly available. Furthermore, the datasets used do not contain any personally identifiable information.

### F.1.2   Robust SSL scenario

We used CIFAR10, MNIST, to demonstrate the effectiveness and stability of RETRIEVE in the robust SSL scenario. The descriptions of the datasets used along with the licenses are given in the Table 1. Both CIFAR10 and MNIST datasets are publicly available. Furthermore, the datasets used do not contain any personally identifiable information.

| Name | No. of classes for ID | No. of classes for OOD | No. samples for labeled | No. samples for unlabeled | No. samples for validation | No. samples for testing | No. of features |
|------|-----------------------|------------------------|-------------------------|---------------------------|----------------------------|-------------------------|-----------------|
| CIFAR10 | 6 | 4 | 2,400 | 20,000 | 5,000 | 10,000 | 32x32x3 |
| MNIST | 6 | 4 | 60 | 30,000 | 10,000 | 10,000 | 28x28x1 |

Table 3: Description of the datasets for robust SSL OOD scenario

| Name | Imbalanced classes | balanced classes | No. samples for labeled | No. samples for unlabeled | No. samples for validation | No. samples for testing | No. of features |
|---|---|---|---|---|---|---|---|
| CIFAR10 | 1-5 | 6-10 | 2,400 | 20,000 | 5,000 | 10,000 | 32x32x3 |

Table 4: Description of the datasets for robust SSL imbalanced scenario

## F.2 Traditional SSL baselines

In this setting, we run RETRIEVE (and all baselines) with warm-start. We incorporate RETRIEVE with three representative SSL methods, including Mean Teacher (MT) [56], Virtual Adversarial Training (VAT) [42] and FixMatch [53]. The baselines considered are RANDOM (where we just randomly select a subset of unlabeled data points of the same size as RETRIEVE), CRAIG [39, 23] and FULL-EARLYSTOP. CRAIG [39, 23] was actually proposed in the supervised learning scenario. We adapt it to SSL by choosing a representative subset of unlabeled points such that the gradients are similar to the unlabeled loss gradients. For more information on the formulation of CRAIG in the SSL case, see Appendix D. We run the per-batch variant of CRAIG proposed in [23], where we select a subset of mini-batches instead of data instances for efficiency and scalability. Again, we emphasize that RANDOM and CRAIG are run with early stopping for the same duration as RETRIEVE. In FULL-EARLYSTOP baseline, we train the model on the entire unlabeled set for the time taken by RETRIEVE and report the test accuracy.

## F.3 Robust SSL baselines

In this setting, we run RETRIEVE (and all baselines) without warm-start. We incorporate RETRIEVE and other baselines with VAT method. DS3L considers a shallow neural network (also called meta-network) to predict the weights of unlabeled examples and estimate the parameters of the neural network based on a clean labeled set (which could also be the original labeled set) via bi-level optimization. For L2RW method, it directly considers the sample weights are hyperparameter and optimize the hyperparameter via bi-level optimization.

## F.4 Experimental Setup

In our experiments, we implement our approaches RETRIEVE for three representative SSL methods, including Mean Teacher (MT), Virtual Adversarial Training (VAT) and fixmatch. For MT and VAT, we built upon the open-source Pytorch implementation[8]. For fixmatch method, we implemented it based on a open-source Pytorch implementation[9]. For DS3L [17], we implemented it based on the released code [10]. For L2RW [49], we used the open-source Pytorch implementation[11] and adapted it to the SSL settings.

We use a WideResNet-28-2 [60] model and a Nesterov's accelerated SGD optimizer with a learning rate of 0.03, weight decay of 5e-4, the momentum of 0.9, and a cosine annealing [35] learning rate scheduler for all the experiments except with MNIST OOD. For the MNIST OOD experiment, we used a two-layer CNN model consisting of two conv2d layers of dimensions 1x16x3 and 16x32x3, two MaxPool2d layers with size=3, stride=2, padding=1, and a RELU activation function. Finally, for MNIST OOD experiments, the optimizer and learning rate schedulers are the same as given above, while the learning rate used is 0.003.

# G Additional Experiments in Traditional SSL

## G.1 RETRIEVE-WARM VS RETRIEVE

We show that RETRIEVE-WARM is more efficient and effective compared to RETRIEVE in the traditional SSL setting in Figure 8. Hence, in our experiments, we consider the warm variant of RETRIEVE in traditional SSL scenario.

---

[8] https://github.com/perrying/pytorch-consistency-regularization
[9] https://github.com/kekmodel/FixMatch-pytorch
[10] https://github.com/guolz-ml/DS3L
[11] https://github.com/danieltan07/learning-to-reweight-examples

**VAT traditional SSL Results**

| Dataset | Model | Budget(%) / Selection Strategy | Top-1 Test accuracy(%) | | | Model Training time(in hrs) | | |
|---|---|---|---|---|---|---|---|---|
| | | | 10% | 20% | 30% | 10% | 20% | 30% |
| CIFAR10 | Wide-ResNet-28-2 | FULL (skyline for test accuracy) | 87.8 | 87.8 | 87.8 | 30.41 | 30.41 | 30.41 |
| | | RANDOM (skyline for training time) | 81.95 | 84.98 | 85.6 | 3.08 | 6.69 | 9.98 |
| | | CRAIG | 83.2 | 85.3 | 86.8 | 3.54 | 7.19 | 10.14 |
| | | RETRIEVE | 84.0 | 85.9 | 87.02 | 3.50 | 7.06 | 10.27 |
| SVHN | Wide-ResNet-28-2 | FULL (skyline for test accuracy) | 93.62 | 93.62 | 93.62 | 19.17 | 19.17 | 19.17 |
| | | RANDOM (skyline for training time) | 87.86 | 90.12 | 91.24 | 1.98 | 3.94 | 5.65 |
| | | CRAIG | 88.86 | 91.25 | 91.94 | 2.12 | 4.08 | 5.98 |
| | | RETRIEVE | 89.3 | 93.2 | 93.3 | 2.18 | 4.2 | 6.1 |

Table 5: Traditional SSL Results for CIFAR10 and SVHN datasets using VAT algorithm

**Mean-Teacher traditional SSL Results**

| Dataset | Model | Budget(%) / Selection Strategy | Top-1 Test accuracy(%) | | | Model Training time(in hrs) | | |
|---|---|---|---|---|---|---|---|---|
| | | | 10% | 20% | 30% | 10% | 20% | 30% |
| CIFAR10 | Wide-ResNet-28-2 | FULL (skyline for test accuracy) | 86.61 | 86.61 | 86.61 | 30.41 | 30.41 | 30.41 |
| | | RANDOM (skyline for training time) | 83.3 | 84.41 | 84.5 | 3.26 | 6.54 | 9.82 |
| | | CRAIG | 83.57 | 84.68 | 85.48 | 3.33 | 6.7 | 11.08 |
| | | RETRIEVE | 83.66 | 85.05 | 86.59 | 3.58 | 6.92 | 10.55 |
| SVHN | Wide-ResNet-28-2 | FULL (skyline for test accuracy) | 94.35 | 94.35 | 94.35 | 13.75 | 13.75 | 13.75 |
| | | RANDOM (skyline for training time) | 91.48 | 92.79 | 93.15 | 1.52 | 2.66 | 4.11 |
| | | CRAIG | 92.11 | 91.81 | 91.98 | 1.7 | 2.88 | 4.49 |
| | | RETRIEVE | 91.84 | 93.13 | 93.76 | 1.57 | 2.94 | 4.30 |

Table 6: FixMatch traditional SSL Results for CIFAR10 and SVHN datasets using Mean-Teacher algorithm

**Traditional SSL Results**

| Dataset | Model | Budget(%) / Selection Strategy | Top-1 Test accuracy(%) | | | Model Training time(in hrs) | | |
|---|---|---|---|---|---|---|---|---|
| | | | 10% | 20% | 30% | 10% | 20% | 30% |
| CIFAR10 | Wide-ResNet-28-2 | FULL (skyline for test accuracy) | 95.52 | 95.52 | 95.52 | 100 | 100 | 100 |
| | | RANDOM (skyline for training time) | 93.8 | 94.31 | 94.4 | 12.76 | 25.6 | 39.1 |
| | | CRAIG | 93.9 | 94.52 | 94.3 | 13.18 | 26.4 | 40 |
| | | RETRIEVE | 94.6 | 94.8 | 94.83 | 13.14 | 26.2 | 39.8 |

Table 7: Traditional SSL Results for CIFAR10 dataset using FixMatch algorithm

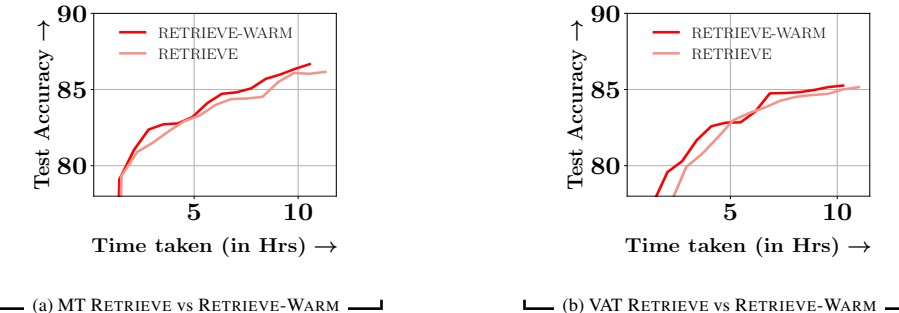

(a) MT RETRIEVE VS RETRIEVE-WARM     (b) VAT RETRIEVE VS RETRIEVE-WARM

Figure 8: Subfigures (a), (b) show comparison of RETRIEVE VS RETRIEVE-WARM with MT, VAT on CIFAR-10 dataset with 30% subset fraction: We show that the RETRIEVE-WARM is more effective and efficient compared to RETRIEVE in traditional SSL setting.

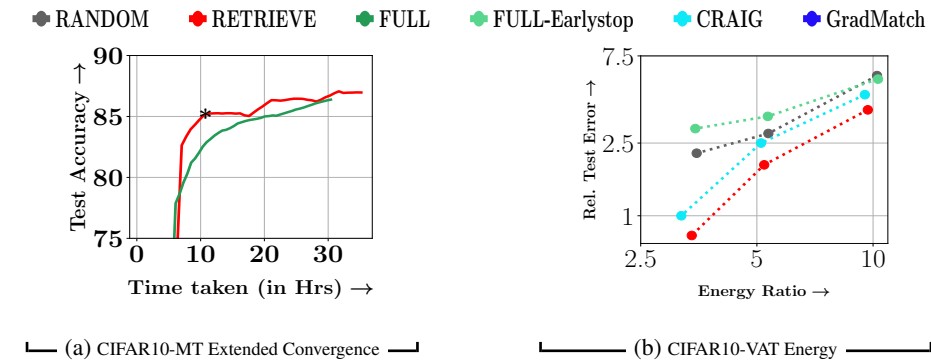

(a) CIFAR10-MT Extended Convergence     (b) CIFAR10-VAT Energy

Figure 7: Subfigure (a) shows MT extended convergence plot on CIFAR-10 dataset with 30% subset fraction: We show that the RETRIEVE achieves similar performance to original MT algorithm while being 1.8X faster and better performance than the original MT algorithm while being 1.5X faster. Subfigure (b) shows the energy efficiency plot of VAT algorithm on CIFAR10 dataset with a subset fractions of 10%, 20%, 30%: We show that the RETRIEVE is 3.1X energy efficient compared to the original VAT algorithm with an accuracy degradation of 0.78%.

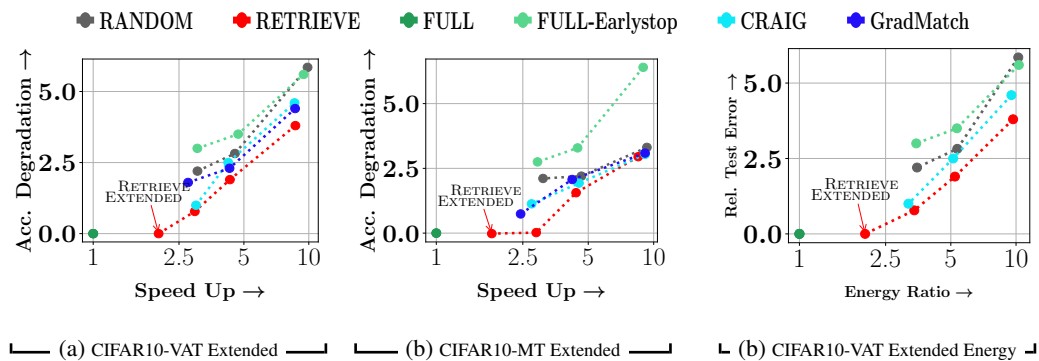

(a) CIFAR10-VAT Extended     (b) CIFAR10-MT Extended     (b) CIFAR10-VAT Extended Energy

Figure 8: Subfigure (a) shows scatter plot of VAT for CIFAR10 dataset with subset fractions of 10%, 20%, 30% along with RETRIEVE-EXTENDED by training the model for more iterations: We show that the RETRIEVE achieves similar performance to original VAT algorithm while being 2X faster. Subfigure (b) shows scatter plot of MT for CIFAR10 dataset with subset fractions of 10%, 20%, 30% along with RETRIEVE-EXTENDED by training the model for more iterations: We show that the RETRIEVE achieves similar performance to original MT algorithm while being 1.8X faster. Subfigure (c) shows energy efficiency scatter plot of VAT for CIFAR10 dataset with subset fractions of 10%, 20%, 30% along with RETRIEVE-EXTENDED by training the model for more iterations: We show that the RETRIEVE achieves similar performance to original VAT algorithm while being 2X more energy efficient.

| VAT Standard Deviation Results | | | | | |
|---|---|---|---|---|---|
| | | | Standard deviation of the Model(for 3 runs) | | |
| | | Budget(%) | 10% | 20% | 30% |
| Dataset | Model | Selection Strategy | | | |
| CIFAR10 | Wide-ResNet-28-2 | FULL | 0.124 | 0.124 | 0.124 |
| | | RANDOM | 0.526 | 0.538 | 0.512 |
| | | CRAIG | 0.368 | 0.285 | 0.195 |
| | | RETRIEVE | 0.198 | 0.148 | 0.105 |
| SVHN | Wide-ResNet-28-2 | FULL | 0.114 | 0.114 | 0.114 |
| | | RANDOM | 0.372 | 0.358 | 0.348 |
| | | CRAIG | 0.284 | 0.241 | 0.207 |
| | | RETRIEVE | 0.187 | 0.154 | 0.112 |

Table 8: Standard deviation results using VAT in traditional SSL scenario for CIFAR10, SVHN datasets for three runs

| Mean-Teacher Standard Deviation Results | | | | | |
|---|---|---|---|---|---|
| | | | Standard deviation of the Model(for 3 runs) | | |
| | | Budget(%) | 10% | 20% | 30% |
| Dataset | Model | Selection Strategy | | | |
| CIFAR10 | Wide-ResNet-28-2 | FULL | 0.105 | 0.105 | 0.105 |
| | | RANDOM | 0.578 | 0.524 | 0.564 |
| | | CRAIG | 0.482 | 0.386 | 0.324 |
| | | RETRIEVE | 0.196 | 0.162 | 0.121 |
| SVHN | Wide-ResNet-28-2 | FULL | 0.11 | 0.11 | 0.11 |
| | | RANDOM | 0.374 | 0.329 | 0.354 |
| | | CRAIG | 0.268 | 0.284 | 0.245 |
| | | RETRIEVE | 0.146 | 0.094 | 0.078 |

Table 9: Standard deviation results using Mean-Teacher in traditional SSL scenario for CIFAR10, SVHN datasets for three runs

| FixMatch Standard Deviation Results | | | | | |
|---|---|---|---|---|---|
| | | | Standard deviation of the Model(for 3 runs) | | |
| | | Budget(%) | 10% | 20% | 30% |
| Dataset | Model | Selection Strategy | | | |
| CIFAR10 | Wide-ResNet-28-2 | FULL | 0.12 | 0.12 | 0.12 |
| | | RANDOM | 0.523 | 0.618 | 0.584 |
| | | CRAIG | 0.386 | 0.342 | 0.305 |
| | | RETRIEVE | 0.174 | 0.142 | 0.105 |

Table 10: Standard deviation results using FixMatch in traditional SSL scenario for CIFAR10, SVHN datasets for three runs

## G.2 Test-Accuracies, Training times and Standard deviations

Table 5, Table 6, Table 7 shows the top-1 test accuracies and training times taken by RETRIEVE and the other baselines considered in traditional SSL scenario for VAT, Mean-Teacher on CIFAR10, SVHN datasets and for FixMatch algorithm on CIFAR dataset for different fractions of 10%, 20% and 30% respectively. Furthermore, Table 8, Table 9, Table 10 gives the standard deviation numbers of RETRIEVE and other baselines in traditional SSL scenarios for VAT, Mean-Teacher on CIFAR10, SVHN datasets and for FixMatch algorithm on CIFAR dataset for different fractions of 10%, 20% and 30% respectively.

## G.3 MT extended convergence plot

Subfigure 9a shows the extended convergence plot of RETRIEVE using Mean-Teacher algorithm on CIFAR10 dataset for 30% subset fraction. From the plot, it is evident that RETRIEVE achieves similar performance to original MT algorithm while being 1.8X faster and better performance than the original MT algorithm while being 1.5X faster.

## G.4 Energy savings

Subfigure 9b shows the energy efficiency plot of RETRIEVE using VAT algorithm on CIFAR10 dataset for 10%, 20%, 30% subset fractions. For calculating the energy consumed by the GPU/CPU cores, we use pyJoules[12]. From the plot, it is evident that RETRIEVE is 3.1X energy efficient compared to the original VAT algorithm with an accuracy degradation of 0.78%.

| RETRIEVE vs RETRIEVE-WARM in Robust SSL | | | | |
|---|---|---|---|---|
| | | | Top-1 Test accuracy(%) | Model Training time(in hrs) |
| | | OOD ratio(%) | 50% | 50% |
| Dataset | Model | Selection Strategy | | |
| CIFAR10 OOD | Wide-ResNet-28-2 | RETRIEVE-WARM | 78.8 | 18.2 |
| | | RETRIEVE | 79 | 17.03 |
| MNIST OOD | Two layer CNN model | RETRIEVE-WARM | 95.3 | 1.43 |
| | | RETRIEVE | 95.85 | 1.37 |
| | | Class imbalance ratio(%) | 50% | 50% |
| CIFAR10 Imbalance | Wide-ResNet-28-2 | RETRIEVE-WARM | 76.13 | 18.6 |
| | | RETRIEVE | 78.86 | 17.3 |

Table 11: RETRIEVE vs RETRIEVE-WARM in Robust SSL scenario for CIFAR10 OOD, MNIST OOD and CIFAR10 Imbalance datasets using VAT algorithm

# H Additional Experiments for Robust SSL

## H.1 RETRIEVE-WARM vs RETRIEVE

We show that RETRIEVE is more efficient and effective compared to RETRIEVE-WARM in the robust SSL setting from the results given in Table 11. For specific numbers, RETRIEVE achieves 79% accuracy in 17.03 hrs while RETRIEVE-WARM achieves 78.8% accuracy in 18.2 hrs for CIFAR10 OOD dataset with an OOD ratio of 50%. Further, RETRIEVE achieves 95.85% accuracy in 1.37 hrs while RETRIEVE-WARM achieves 95.3% accuracy in 1.43 hrs for MNIST OOD dataset with an OOD ratio of 50%. Finally, RETRIEVE achieves 78.86% accuracy in 17.3 hrs while RETRIEVE-WARM achieves 76.13% accuracy in 18.6 hrs for CIFAR10 Imbalance dataset with a class imbalance ratio of 50%. Hence, in our experiments, we consider RETRIEVE without warm variant in robust SSL scenario.

## H.2 Test-Accuracies, Training times and Standard deviations:

Table 12 shows the top-1 test accuracies and training times taken by RETRIEVE and the other baselines considered in robust SSL scenario for VAT on CIFAR10 OOD, MNIST OOD, and CIFAR10 Imbalance datasets. The results show that RETRIEVE with VAT outperforms all other baselines, including DS3L [17] (also run with VAT) in the class imbalance scenario as well. In particular, RETRIEVE outperforms other baselines by around 1.5% on the CIFAR-10 with imbalance. Furthermore, Table 13 gives the standard deviation numbers of RETRIEVE and other baselines in robust SSL scenario for VAT on CIFAR10 OOD, MNIST OOD, and CIFAR10 Imbalance datasets.

# I Broader Impacts and Limitations

**Limitations:** One of the main limitations of RETRIEVE is that even though it reduces the training time, energy costs, and CO2 emissions of SSL algorithms, it does not reduce the memory requirement. Furthermore, the memory requirement is a little higher because it requires additional memory to store the gradients required for the coreset selection, which makes running the RETRIEVE algorithm in devices with low memory capacity significantly harder without proper memory handling.

**Societal Impacts:** We believe RETRIEVE has a significant positive societal impact by making SSL algorithms (and specifically robust SSL) significantly faster and energy-efficient, thereby reducing the CO2 emissions and energy consumption incurred during training. This is particularly important because state-of-the-art SSL approaches like FixMatch are very computationally expensive. Furthermore, SSL approaches often have a large number of hyper-parameters and the performance can be

---

[12]`https://pypi.org/project/pyJoules/`.

VAT Robust SSL Results

| Dataset | Model | Selection Strategy | Top-1 Test accuracy(%) | | | Model Training time(in hrs) | | |
|---|---|---|---|---|---|---|---|---|
| | | OOD ratio(%) | 25% | 50% | 75% | 25% | 50% | 75% |
| CIFAR10 OOD | Wide-ResNet-28-2 | VAT | 76.3 | 75.6 | 74.25 | 30.34 | 30.34 | 30.34 |
| | | SUPERVISED | 76.1 | 76.1 | 76.1 | 0.22 | 0.22 | 0.22 |
| | | L2RW | 78.2 | 75.5 | 73.4 | 86.51 | 86.58 | 86.62 |
| | | DS3L | 78.8 | 77.6 | 76.3 | 88.94 | 88.91 | 88.92 |
| | | RETRIEVE | 79.26 | 79 | 76.56 | 17.36 | 17.03 | 17.12 |
| MNIST OOD | Two layer CNN model | VAT | 95 | 92.2 | 88.1 | 2.46 | 2.46 | 2.46 |
| | | SUPERVISED | 93 | 93 | 93 | 0.01 | 0.01 | 0.01 |
| | | L2RW | 95.2 | 88.5 | 87.5 | 7.34 | 7.29 | 7.23 |
| | | DS3L | 97.1 | 95.8 | 92.1 | 7.22 | 7.18 | 7.12 |
| | | RETRIEVE | 97.3 | 95.85 | 93.48 | 1.365 | 1.37 | 1.36 |
| | | Class imbalance ratio(%) | 10% | 30% | 50% | 10% | 30% | 50% |
| CIFAR10 Imbalance | Wide-ResNet-28-2 | VAT | 56.12 | 65.15 | 72.14 | 30.24 | 30.26 | 30.2 |
| | | SUPERVISED | 58.12 | 64.21 | 71.12 | 0.22 | 0.22 | 0.22 |
| | | L2RW | 61.54 | 68.45 | 71.24 | 87.35 | 87.19 | 87.41 |
| | | DS3L | 63.54 | 73.89 | 77.41 | 88.16 | 88.04 | 88.5 |
| | | RETRIEVE | 66.88 | 75.83 | 78.86 | 17.27 | 17.31 | 17.3 |

Table 12: Robust SSL Results for CIFAR10 OOD, MNIST OOD and CIFAR10 Imbalance datasets using VAT algorithm

| | | | Standard deviation of the Model(for 3 runs) | | |
|---|---|---|---|---|---|
| | | VAT Standard Deviation Results | | | |
| | | OOD ratio(%) | 25% | 50% | 75% |
| Dataset | Model | Selection Strategy | | | |
| CIFAR10 OOD | Wide-ResNet-28-2 | VAT | 0.13 | 0.18 | 0.24 |
| | | SUPERVISED | 0.021 | 0.021 | 0.021 |
| | | L2RW | 0.31 | 0.39 | 0.295 |
| | | DS3L | 0.38 | 0.41 | 0.34 |
| | | RETRIEVE | 0.26 | 0.21 | 0.27 |
| MNIST OOD | Two layer CNN model | VAT | 0.014 | 0.018 | 0.021 |
| | | SUPERVISED | 0.01 | 0.01 | 0.01 |
| | | L2RW | 0.04 | 0.03 | 0.04 |
| | | DS3L | 0.061 | 0.041 | 0.056 |
| | | RETRIEVE | 0.034 | 0.039 | 0.036 |
| | | Class imbalance ratio(%) | 10% | 30% | 50% |
| CIFAR10 Imbalance | Wide-ResNet-28-2 | VAT | 0.295 | 0.242 | 0.185 |
| | | SUPERVISED | 0.16 | 0.13 | 0.11 |
| | | L2RW | 0.37 | 0.32 | 0.26 |
| | | DS3L | 0.34 | 0.36 | 0.21 |
| | | RETRIEVE | 0.32 | 0.28 | 0.205 |

Table 13: Standard deviation results using VAT in Robust SSL scenario for CIFAR10 OOD, MNIST OOD and CIFAR10 Imbalance datasets for three runs.

heavily dependent on the right tuning of these hyper-parameters [53, 44]. We believe that RETRIEVE can enable much faster and energy efficient tunings of hyper-parameters in SSL approaches thereby enabling orders of magnitude speedup and CO2 emissions being reduced. RETRIEVE takes one step towards **Green-AI** by enabling using smaller subsets for training these models.