# OpenReview forum: "RETRIEVE: Coreset Selection for Efficient and Robust Semi-Supervised Learning"
_NeurIPS.cc/2021/Conference — NeurIPS 2021 Poster_

### Official Review · Reviewer_3kHQ · 2021-07-12

**Rating:** 6
**Confidence:** 3

**Summary:**

This paper proposes RETRIEVE to address the computational cost issue in previous semi-supervised learning (SSL) algorithms. The key idea is to train SSL on a subset of unlabeled data (coreset) instead of the entire unlabeled data. To do so, it must solve the problem of selecting the coreset efficiently and effectively. RETRIEVE tackles this problem by solving a two-level optimization problem such that the selected coreset minimizes the labeled set loss. This paper evaluates RETRIEVE on two settings---traditional SSL and robust SSL---and demonstrates its improvement in terms of speed, energy consumption, and CO2 emissions compared to prior work.

**Limitations And Societal Impact:**

This paper does not discuss any limitation. I would recommend to add limitations in the revision, which definitely exist. The paper mentions positive societabl impact in terms of less energy consumption and CO2 emissions during training, which is good.


**Main Review:**

**Originality**: The proposed method is new. The novelty of RETRIEVE is to train on coreset of unlabeled data rather than on the entire unlabeled data. The challenge is then how to select the high-quality coreset effectively and efficiently. They use greedy selection and Taylor approximation. Generally speaking, the novelty of this work is incremental:
- The subset selection problem is NP-hard in general [1].
- The connection with submodular optimization has been demonstrated before [2].

[1] Balas Kausik Natarajan. Sparse approximate solutions to linear systems. SIAM journal on computing, 24(2):227–234, 1995.

[2] Baharan Mirzasoleiman, Jeff Bilmes, and Jure Leskovec. Coresets for data-efficient training of machine learning models. In International Conference on Machine Learning, pages 6950–6960. PMLR, 2020.

**Quality**: The submission is technically sound. The claims in the contribution list are well supported by theoretical analyses and empirically results.  I like the extra measurement in terms of energy consumption and CO2 emissions. The related work is adequately cited.

**Clarity**: This paper is well-written and easy to follow. The problem is also well-motivated from the introduction by asking two questions. The experimental details are also very specific such that reproducing the results should be possible.

**Significance**: Empirically, RETRIEVE outperforms prior work in two different settings. I would recommend the authors to add releasing code link in the revision.

Here are a few questions:
- Can RETRIEVE generalize to machine learning methods other than deep neural network? For instance, logistic regression, graident boosting tree?
- If no to the above question, would it be a limitation of this work?




**Time Spent Reviewing:**

2

---

> ### Author Response · Authors · 2021-08-10
> **Response to Reviewer - 3kHQ**
>
> We thank you for your detailed feedback. We will address minor issues, e.g., typos, grammatical errors, notations, etc., in the next version.
>
> ### Originality
> > **Q1. The subset selection problem is NP-hard in general [1].**
>
> The proof of NP-hardness of subset selection problem depends on the objective function in hand, and hence the proof given in [1], which is proved explicitly for sparse approximation of linear systems, does not hold for the optimization problem we consider in **Equation 5 given in  the main paper**
>
> > **Q2. The connection with submodular optimization has been demonstrated before [2].**
>
> Similarly, the connection of coreset selection with submodular optimization given in CRAIG [2] does not prove that RETRIEVE’s set function is submodular as the respective set functions used in RETRIEVE and CRAIG are different.
>
> > **Q3. Code Link**
>
> We added the link to code and licenses of the code for RETRIEVE in **section A.1 of the appendix.**
>
> > **Q4. Can RETRIEVE generalize to machine learning methods other than deep neural networks? For instance, logistic regression, gradient boosting tree?**
>
> RETRIEVE can be generalized to machine learning methods as long as they use a loss function that can be solved by gradient descent. Hence, RETRIEVE can be used for logistic regression, regression cases as well. RETRIEVE cannot be extended to gradient boosting trees as the tree structures cannot be solved by gradient descent and instead use an additive strategy. Hence, we need to develop specialized coreset selection methods for gradient boosting trees which are out of scope for our current work.
>
> > **Q5. If no to the above question, would it be a limitation of this work?**
>
>   However, we would not consider it a limitation as RETRIEVE can be widely extended to machine learning methods that use loss-based objectives that can be solved by gradient descent. Furthermore, the main goal of RETRIEVE is to reduce the computational cost of deep SSL, something which is significant for state-of-the-art deep SSL approaches like FixMatch. The use of something like RETRIEVE may not be as beneficial in a learning paradigm like linear or logistic regression, which will already be fast.

---

### Official Review · Reviewer_k5XM · 2021-07-16

**Rating:** 7
**Confidence:** 4

**Summary:**

This work extends previous approaches for coreset selection to the problem of semi-supervised learning (SSL). Experimental results show the approach has good performance even in settings with data imbalance and out of domain training data


**Limitations And Societal Impact:**

Limited discussion of limitations/societal impact

**Main Review:**

This work extends previous approaches for coreset selection to the problem of semi-supervised learning (SSL). Experimental results show the approach has good performance even in settings with data imbalance and out of domain training data

Originality/Significance: Straightforward application of previous ideas from methods like CRAIG and GLISTER to the problem of SSL. This may have impact in the SSL community but I think it's significance for data summarization/selection is rather limited.

Clarity: Well organized overall, good review of related work that unifies SSL algorithms under a common framework and distinguished the proposed approach from previous SSL methods

Quality/Methodology: Proposed method appears to improve consistently over previous work, but authors should compare against GLISTER and GRAD-MATCH

Questions:
- How would the method perform on data from non-image domains?

---
EDIT: The authors fully addressed my main questions about additional baselines and problem domains. Therefore I am increasing my score to 7

**Time Spent Reviewing:**

4

---

> ### Author Response · Authors · 2021-08-10
> **Response to Reviewer-k5XM**
>
> We thank you for your detailed feedback. We will address minor issues, e.g., typos, grammatical errors, notations, etc., in the next version.
>
> >**Q1: Proposed method appears to improve consistently over previous work, but authors should compare against GLISTER and GRAD-MATCH**
>
> GLISTER and GRAD-MATCH are both specialized for supervised learning, and can not be directly applied to SSL. Our proposed method RETRIEVE is an extension of GLISTER for efficient and robust SSL.  GLISTER, RETRIEVE, and Safe-SSL approaches all come under the family of bi-level optimization approaches; however, in each case, we need to adapt it to the particular scenario appropriately. On the other hand, Grad-Match and CRAIG find coresets or data subsets by finding representative points that approximate the gradient of the loss function (in this case, the SSL loss on the unlabeled set).
>
> To make the empirical results stronger, in the rebuttal phase, we added two additional baselines -- one is Grad-Match[1] and the other is Bayesian coresets[2] both adapted to the SSL setting (i.e., where we construct a coreset based on either the unlabeled loss gradients in grad-match and unlabeled set logits in Bayesian-Coreset) on the CIFAR10 dataset, using Mean Teacher as SSL algorithm and Wide-ResNet-28-2 model in traditional SSL setting. Below are the results:
>
> ### Results:
> - Dataset: CIFAR10
> - Model: Wide-ResNet-28-2
> - SSL Algorithm: Mean Teacher
>
> >Budget: 30%
>
> | Selection Strategy | Test Accuracy| Training time (in Hrs)|
> |-----------------------|-----------------|-----------------|
> |Bayesian Coreset|85.67%|16.451|
> |GradMatch|85.87% |12.451|
> |RETRIEVE|**86.59%**|**10.55**|
>
> >Budget: 20%
>
> | Selection Strategy | Test Accuracy| Training time (in Hrs)|
> |-----------------------|-----------------|-----------------|
> |Bayesian Coreset|84.18%|10.51|
> |GradMatch|84.54%|7.21|
> |RETRIEVE |**85.05%** |**6.92**|
>
> The results show that RETRIEVE achieves better performance than the extensions of both Bayesian coreset selection method and GradMatch algorithm in terms of model performance and speedup. Even though Grad-Match works in the supervised setting, it doesn't seem to work well in the SSL case. We believe this is probably because of a) the difficulty in matching gradients in more complex SSL losses, and b) the fact that the predictions and sample losses on the unlabeled set are noisy because of the usage of hypothesized label loss gradients, and hence weighted subsets may not work as well. On the other hand, the Bi-level formulation of RETRIEVE works very well; the success of bi-level formulation can also be seen in continuous bi-level approaches like Safe-SSL for robust learning, albeit those come at a higher computational cost. Even though the Bayesian Coreset approach worked similarly to GradMatch in terms of accuracy, it is significantly slower. This is due to the fact that the Bayesian coreset approach was not developed for efficient learning but instead was developed to capture coresets that try to represent the log-likelihood of the entire dataset that MCMC methods can further use. We will add these results to the paper.
>
> >**Q2: How would the method perform on data from non-image domains?**
>
> We performed an additional experiment on non-image domain datasets, and the results are given below. Specifically, we considered a sentiment analysis task on the IMDB dataset using LSTM based model and VAT algorithm. We followed the experimental settings given in the following paper[3].
>
> ### Results:
>
> -   Dataset: IMDB
> -   Model: LSTM based Neural Network
> -   SSL algorithm: VAT
>
> > Budget: 30%
>
> |Selection Strategy |Test Accuracy| Training Time |
> |-------------------|---------------|---------------|
> |Full Training(Skyline for accuracy)| 94.1%|0.78|
> |Random(Skyline for Time)|91.3%|0.25|
> |CRAIG|93.1%|0.31|
> |RETRIEVE|**93.6%**| **0.29**|
> |Full EarlyStop|92.8%|0.30|
>
> The results show that RETRIEVE achieves a 2.68x speedup compared to full dataset training while only losing 0.5% accuracy using a 30% subset. Hence, the results show that RETRIEVE performs well on Non-Image domains as well.
>
> ### References:
> [1]. Killamsetty, K., S, D., Ramakrishnan, G., De, A. &amp; Iyer, R.. (2021). GRAD-MATCH: Gradient Matching based Data Subset Selection for Efficient Deep Model Training. _Proceedings of the 38th International Conference on Machine Learning_, in _Proceedings of Machine Learning Research_ 139:5464-5474 Available from http://proceedings.mlr.press/v139/killamsetty21a.html.
>
> [2]. Campbell, T. &amp; Broderick, T.. (2018). Bayesian Coreset Construction via Greedy Iterative Geodesic Ascent. _Proceedings of the 35th International Conference on Machine Learning_, in _Proceedings of Machine Learning Research_ 80:698-706 Available from http://proceedings.mlr.press/v80/campbell18a.html.
>
> [3]. Miyato, T., Dai, A.M., & Goodfellow, I. (2016). Virtual Adversarial Training for Semi-Supervised Text Classification. ArXiv, abs/1605.07725.

---

### Official Review · Reviewer_rNnr · 2021-07-16

**Rating:** 7
**Confidence:** 2

**Summary:**

In this paper the authors propose a framework to select a subset of unlabeled data for semisupervised learning using a greedy selection approach.  The authors show the effectiveness of their method by extensive experimentaion on real world data sets. They show that their method gives significant speedups in traditional semi supervised learning tasks with very less drop in accuracy. For Robust SSL where the unlabeled data has out of distribution points or class imbalance , their method actually outperforms the state of the art with a significant speedup.

**Limitations And Societal Impact:**

For limitations please refer to the above section.

I donot see any major potential negative impact of this work.

For improvement please highlight the technical challenges in theory and your theoretical contributions more in writing. Also would be good to give more details (related work) about traditional coresets and contrast your coresets with traditional ones.

**Main Review:**

The paper proposes a framework to select a subset of unlabeled data for semisupervised learning. The subset selection and the learning task are done together by solving a bilevel discrete-continuous optimization problem. The authors show that the task of subset selection can be done using a greedy approach as the function is weakly submodular. They demonstrate the effectiveness of the framework with extensive experimentation

Significance:
The task of labeling data is very costly. Hence semisupervised learning is important . However the state of art algorithms for SSL are expensive and for Robust SSL even more so. Hence the problems addressed in the paper are well motivated.

Originality:
From the theoretical point of view, the paper is not that novel. The main theoretical contribution, I believe, is the framework itself and the proposal of the subset selection problem along with learning as a bilevel optimization and proving it weakly submodular. The proof of theorem builds up on existing techniques. Also traditionally the word "coreset" is used when there are some provable guarantees(in terms of sample size, approximation etc.). However in this paper either there are no such guarantees or at least they are not clearly highlighted in the writing.
That being said , the paper, does a very good task in the experimentation. The experiments are done on large real world datasets and also the subset is used with atleast 3 state of the art techniques. Most of the details and setup is stated clearly. The subset is compared with random sample, early stopping  and also CRAIG (which was proposed for supervised learning. Modifying that for SSL is also a minor contribution). Advantage is shown not only in terms of convergence time but also in tems of CO_2 emission.  Experiments for Robust SSL beat the state of art method both in terms of time and accuracy.

Clarity and Quality:
I didnot go through the proof in details but it looks ok. I would suggest to highlight the theoretical contributions more in the writing and make them clear. Also in writing there is sudden change in tense (from present to past tense) on lines 266 and 268. In terms of empirical results this is a very good paper and will be of interest to ML community in general and specifically to the ML practitioners and ML engineers. Also the appendices are quite detailed and give a good overview of state of art SSL algorithms.

Question: Just as the authors have modified CRAIG for the semisupervised setting, is it possible to modify the CRUST algorithm used to work with nosiy labels ( Baharam et al Neurips 2020. ) for Robust SSL?

**Time Spent Reviewing:**

5

---

> ### Author Response · Authors · 2021-08-10
> **Response to Reviewer - rNnr**
>
> We thank you for your detailed feedback. We will address minor issues, e.g., typos, grammatical errors, notations, etc., in the next version.
>
> > **Q1. Just as the authors have modified CRAIG for the semisupervised setting, is it possible to modify the CRUST algorithm used to work with noisy labels ( Baharan et al. Neurips 2020. ) for Robust SSL?**
>
> CRUST is a coreset selection method developed in a supervised learning setting to tackle noisy labels. Hence, it has limited applicability in the SSL scenario where we are trying to select a coreset of an unlabeled set for which the labels are not available. However, we can modify the CRUST algorithm by considering the last layer gradients on SSL loss and apply it in the Robust SSL setting. Hence, we ran some additional experiments in the Robust SSL setting on the CIFAR10 OOD dataset using Wide-ResNet-28-2 model and VAT as SSL algorithm for OOD ratios of 50% and 75% with CRUST, and the results were given below:
>
> ### Results
>
> -   Dataset: CIFAR10 OOD
> -   Model: Wide-ResNet-28-2
> -   SSL Algorithm: VAT
>
> > OOD Ratio: 50%
>
> |Selection Strategy|Test Accuracy|Training time (in Hrs)|
> |-------------------|--------------|----------------------|
> |CRUST|76.71%|17.62|
> |RETRIEVE|**79%**|**17.03**|
>
> > OOD Ratio: 75%
>
> |Selection Strategy | Test Accuracy | Training time (in Hrs) |
> |-------------------|--------------|----------------------|
> |CRUST| 75.21%| 17.58|
> |RETRIEVE|**76.56%**| **17.12**|
>
>   The results show that CRUST didn’t perform well in terms of accuracy and speedups achieved compared to RETRIEVE. Except for MixUP, CRUST is similar to CRAIG, which didn’t perform well compared to RETRIEVE in a traditional SSL setting. Furthermore, the performance gain due to MixUP for coreset selection in the SSL setting is minimal. The minimal gain can be attributed to the fact that the hypothesized labels used for MixUP in the earlier stages of training are noisy. Furthermore, as stated above, CRUST was developed to tackle noisy labels in a supervised learning setting and is not developed to deal with OOD or Class Imbalance in general.

---

> > ### Comment · Reviewer_rNnr · 2021-08-25
> > **after rebuttal**
> >
> > thanks for the detailed response and the additional experiments. I strongly encourage you to highlight the technical challenges and guarantees in the final version of the paper. I maintain my score of 7.

---

### Official Review · Reviewer_fgXZ · 2021-07-21

**Rating:** 6
**Confidence:** 3

**Summary:**

In this paper, the authors propose a coreset selection framework, called RETRIEVE, that aims to improve the efficiency and robustness of semi-supervised learning (SSL).
The main insight that underlies RETRIEVE is that effective selection of a subset of unlabeled data - instead of using the entire unlabeled data - and training on such a carefully selected subset can enable current SSL algorithms to converge faster and significantly reduce the computational cost of the learning process.
The study empirically demonstrates that RETRIEVE can reduce the training time and improve the performance for out-of-distribution or imbalanced data for various SSL algorithms.


**Limitations And Societal Impact:**

The authors discuss limitations of the current work and also discuss any potential negative societal impact in the supplementary material.


**Main Review:**

GENERAL COMMENTS

This study presents the potential benefits of employing coreset selection schemes in the context of semi-supervised learning.
Evaluation results based on a number of benchmarks show that the proposed coreset selection scheme, RETRIEVE, may potentially improve the performance of various SSL algorithms - especially, for out-of-distribution (OOD) data and imbalanced datasets.
However, the evaluation results are somewhat limited and further evaluations of RETRIEVE, as well as comparisons with other coreset selection schemes, would be needed to demonstrate the benefits of RETRIEVE in a more convincing manner.
There are also a number of ambiguities in the paper that need to be clarified.


DETAILED COMMENTS

1. Figure 1 is not very informative in its current form.
The authors may want to consider removing it and instead provide a better explanation in the main text or try revising the figure to make it more informative.

2. "Accuracy degradation" is not clearly defined and its use is a bit confusing.
For example, when the accuracy is improved (despite the speedup), it is shown as "negative" accuracy degradation.
It would be better to show the "accuracy" or "accuracy improvement" rather than "accuracy degradation" to make it easier to understand.

3. In the caption for Figure 2, does "Efficient SSL" refer to "Traditional SSL"?
In other places of the paper, the authors typically refer to "Traditional SSL" and "Robust SSL", and it would be better to use consistent terminology throughout the paper.

4. In lines 77-79, what does it mean that RETRIEVE "minimizes the performance on the labeled dataset"?
Please rephrase to make the meaning clear.

5. It is unclear what is meant by "Equation (5) can be solved efficiently using greedy algorithms [30, 31] with approximation guarantees."
What are the mentioned "approximation guarantees"?
This requires further elaboration.

6. In the performance evaluation results, only Craig was tested and compared with the proposed coreset section scheme.
To demonstrate the advantages of RETRIEVE, performance should be evaluated for other combinations of recent coreset selection algorithms (e.g., Bayesian coreset as well as other coreset selection schemes discussed in lines 129-137) with SSL.

7. It is mentioned (lines 287-289) that "for the Robust SSL scenario, we do not warm start with the full unlabeled set for training" and that this is "because training on an entire unlabeled set containing biases which instead hurts the model performance"
Please provide further insights underlying this statement.
Also, this sentence appears to be broken and needs to be rephrased.

8. Please clearly explain how the speed-up and the accuracy degradation were evaluated.
Furthermore, explain what the data points shown in the figures represent.

9. Despite the importance in the proposed greedy algorithm, connections with weak-submodularity and its implications are not discussed in detail (e.g., lines 366-367)
Please elaborate.

10. Is there a way to balance the trade-off between speed and accuracy when integrating RETRIEVE into SSL algorithms?


----------------------------

NOTE:

The original rating has been updated after reviewing the authors' point-by-point response to the above comments.




**Time Spent Reviewing:**

10

---

> ### Author Response · Authors · 2021-08-10
> **Response to Reviewer -fgXZ**
>
> We thank you for your detailed feedback. We will address minor issues, e.g., typos, notations, etc., in the next version.
>
>   > **Q1. Figure 1 is not very informative in its current form.**
>
> We will revise the figure to make it more informative in the next version. Figure 1 visually illustrates three different kinds of SSL scenarios that we considered in the paper. Figure 1(a) shows an example of an unlabeled dataset with the same distribution as the labeled set, Figure 1(b) shows an example of an unlabeled dataset containing OOD examples that are not in the labeled set, and Figure 1(c) shows an example of an unlabeled dataset with class imbalance.
>
> > **Q2. "Accuracy degradation" is not clearly defined and its use is a bit confusing.**
>
> We agree that the usage of the term "Accuracy degradation" is confusing, especially when "RETRIEVE" is performing better than full training, especially in the robust SSL setting. To avoid confusion, We will change it to "Relative Accuracy difference" and define it clearly -- note that this relative accuracy difference will be computed with respect to the model using all the data.
>
>  > **Q3. In the caption for Figure 2, does "Efficient SSL" refer to "Traditional SSL"?**
>
> The term "Efficient SSL" used in Figure 2 refers to the "Traditional SSL" setting. We will change the term to "Traditional SSL" in the next version.
>
> > **Q4. In lines 77-79, what does it mean that RETRIEVE "minimizes the performance on the labeled dataset"? Please rephrase to make the meaning clear.**
>
> We apologize for the typo in lines 77-79 that we realized after submission. We will rephrase the sentence in the next version. The actual sentence is that "RETRIEVE selects a coreset of the unlabeled set, which, when trained using the combination of the labeled set and the specific unlabeled data coreset, minimizes the model's loss on the labeled dataset."
>
> > **Q5. It is unclear what is meant by "Equation (5) can be solved efficiently using greedy algorithms [30, 31] with approximation guarantees." What are the mentioned "approximation guarantees"? This requires further elaboration.**
>
> Since the optimization problem given in Equation(5) is weakly submodular, we can solve it using greedy algorithms with approximation guarantees. In this context, the greedy algorithm approximation guarantee is a measure of how close the set function value using the subset selected by the greedy algorithm is compared to the set function value using the optimal subset. For example, if the approximation guarantee is 0.5, the greedy subset has a set function value at least 0.5 times the optimal set function value. In our experiments, we used the stochastic greedy selection method, and we mentioned the approximation guarantee achieved by the stochastic greedy selection method **in lines 199-201 of the main paper**.
>
> > **Q6. In the performance evaluation results, only Craig was tested and compared with the proposed coreset section scheme. To demonstrate the advantages of RETRIEVE, performance should be evaluated for other combinations of recent coreset selection algorithms (e.g., Bayesian coreset as well as other coreset selection schemes discussed in lines 129-137) with SSL.**
>
> To demonstrate the advantages of RETRIEVE, we performed additional experiments in traditional SSL setting on the CIFAR10 dataset, using Mean Teacher as SSL algorithm and Wide-ResNet-28-2 model, comparing the performance of RETRIEVE with the GradMATCH [1] and Bayesian coreset selection [2] algorithm.
>
>   ### Results:
> - Dataset: CIFAR10
> - Model: Wide-ResNet-28-2
> - SSL Algorithm: Mean Teacher
>
> >Budget: 30%
>
> | Selection Strategy | Test Accuracy| Training time (in Hrs)|
> |-----------------------|-----------------|-----------------|
> |Bayesian coreset|85.67%|16.451|
> |GradMatch|85.87% |12.451|
> |RETRIEVE|**86.59%**|**10.55**|
>
>  >Budget: 20%
>
> | Selection Strategy | Test Accuracy| Training time (in Hrs)|
> |-----------------------|-----------------|-----------------|
> |Bayesian coreset|84.18%|10.51|
> |GradMatch|84.54%|7.21|
> |RETRIEVE |**85.05%** |**6.92**|
>
>
> The results show that RETRIEVE achieves better performance than both Bayesian coreset and GradMatch algorithm in terms of model performance and speedup. Even though Grad-Match works in the supervised setting, it doesn't seem to work well in the SSL case. We believe this is probably because of a) the difficulty in matching gradients in more complex SSL losses, and b) the fact that the predictions and sample losses on the unlabeled set are noisy because of the usage of hypothesized label loss gradients, and hence weighted subsets may not work as well. On the other hand, the Bi-level formulation of RETRIEVE works very well; the success of bi-level formulation can also be seen in continuous bi-level approaches like Safe-SSL for robust learning, albeit those come at a higher computational cost. Even though the Bayesian Coreset approach worked similarly to GradMatch in terms of accuracy, it is significantly slower. This is due to the fact that the Bayesian coreset approach was not developed for efficient learning but instead was developed to capture coresets that try to represent the log-likelihood of the entire dataset that MCMC methods can further use. We will add these results to the paper.
>
> > **Q7. It is mentioned (lines 287-289) that "for the Robust SSL scenario, we do not warm start with the full unlabeled set for training" and that this is "because training on an entire unlabeled set containing biases instead hurts the model performance" Please provide further insights underlying this statement. Also, this sentence appears to be broken and needs to be rephrased.**
>
> We will address the grammatical errors in the next version of the paper. Our central insight for not using warm start is that when the unlabeled set contains OOD or class Imbalance, the standard full SSL training often performs poorly by the biases introduced in the model due to a distribution mismatch between labeled set 7and unlabeled set. Hence, warm starting the model by full training in such scenarios corresponds to using a biased model for subset selection which may affect the subset selection process and further hurt the performance. Furthermore, **in Table 11 of the Appendix**, we compare the performance of RETRIEVE with warm Start and RETRIEVE w/o warm start, and the results show that RETRIEVE with warm start performs poorly compared to RETRIEVE w/o warm start in Robust SSL setting.
>
> > **Q8. Please clearly explain how the speedup and the accuracy degradation were evaluated. Furthermore, explain what the data points shown in the figures represent.**
>
> "_ Speedup _" achieved for a particular method is defined as the ratio of training time taken by full training and the training time taken by the model on the subsets selected by the method. "_Accuracy Degradation_" for a particular method is defined as the difference between full training model accuracy and model accuracy achieved by training on the subsets selected by the considered method. We will add the detailed definition in the next version of the paper and we will use the term "Relative Accuracy Difference" instead of "Accuracy Degradation".
>
> As explained **in line numbers 326-327**, the different data points shown in the figures correspond to different subset sizes (i.e., 10%, 20%, and 30% of unlabeled set) of the unlabeled set; with the rightmost point corresponding to the smallest subset, i.e., 10% and the leftmost point corresponding to the largest subset, i.e., 30% subset.
>
> > **Q9. Despite the importance in the proposed greedy algorithm, connections with weak-submodularity and its implications are not discussed in detail (e.g., lines 366-367). Please elaborate.**
>
> We used the stochastic greedy selection method because it is significantly faster and has a time complexity of O(n), where n is the unlabeled set size compared to a naive greedy algorithm's time complexity of O(nk), where k is the subset size.  We gave a detailed description of weak submodularity and the proof that the RETRIEVE set function is weakly submodular **in Section B of Appendix.** We also provided the approximation guarantees of subsets selected by the greedy algorithm **in lines 753-758 of the Appendix.** However, we will add a few details in the main paper as well -- thanks for pointing this out!
>
> > **Q10. Is there a way to balance the trade-off between speed and accuracy when integrating RETRIEVE into SSL algorithms?**
>
> Empirically we observed that using subset sizes of 30% often performs comparably to full training in terms of accuracy while achieving a speedup of at least 2.5 X. Hence if the goal is to not lose any accuracy, we would suggest using 30% of the data with RETRIEVE (we observe this consistently across datasets and settings). However, if one is willing to sacrifice a little on the accuracy, one can go even lower at 10% or 20%. Furthermore, suppose one is doing something like the hyper-parameter tuning of SSL algorithms. In that case, the users can use RETRIEVE with even 5% to 10% subset sizes because in such a case, one can sacrifice accuracy in the interest of speedup. To summarize, the trade-off depends on the use case, but we can recommend using RETRIEVE with 30% subsets as a default setting. We will clarify this in the paper -- thanks for pointing this out!
>
> ### References
>
> [1]: Killamsetty, K., S, D., Ramakrishnan, G., De, A. &amp; Iyer, R.. (2021). GRAD-MATCH: Gradient Matching based Data Subset Selection for Efficient Deep Model Training, Proceedings of the 38th International Conference on Machine Learning, Available from http://proceedings.mlr.press/v139/killamsetty21a.html.
>
> [2]: Campbell, T. &amp; Broderick, T.. (2018).. Bayesian Coreset Construction via Greedy Iterative Geodesic Ascent. Proceedings of the 35th International Conference on Machine Learning, Available from http://proceedings.mlr.press/v80/campbell18a.html.

---

> > ### Comment · Reviewer_fgXZ · 2021-08-24
> > **Response to authors' comments**
> >
> > I would like to thank the authors for their detailed point-by-point response to the review comments.
> > The authors have clarified the ambiguities in their original manuscript and addressed many of the concerns pointed out in the review.
> > Furthermore, the performance comparison with two other recent coreset selection methods provides further confidence about the potential advantages of the proposed scheme.
> > As a result, I am updating my rating accordingly.

---

### Public Comment · ~Xilie_Xu1 · 2022-12-05
**Question about proof of \alpha-submodularity**

Dear Authors,

I am interested in the wonderful coreset selection RETRIEVE proposed in this paper. This indeed speeds up semi-supervised training.

However, I have a question regarding proof of $\alpha$-submodularity. In Appendix B.5, $f(S)$ was converted to $\alpha_0\lambda_0kng_{nm}+ \dots$. Then, you remove this term $\alpha_0\lambda_0kng_{nm}$ due to that you regard this term as a constant. But, actually, this term seems to be related to $S$ where $k$ is the size of set $S$. Therefore, it seems that it is incorrect to convert optimization problem $S^* = \arg \max f(S)$ to the optimization problem $S^* = \arg \max \hat{f}(S)$ since $f(S) = \hat{f}(S) + \alpha_0\lambda_0ng_{nm} |S|$.

Really hope to hear clarification about this question. Thank you!

---

> ### Public Comment · ~Krishnateja_Killamsetty1 · 2022-12-05
> **\alpha Submodularity Clarification**
>
> Hello,
>
> Thanks for your interest in our work; this is a great question. In the proof section, we ignored the constant even though it is dependent on subset size $k$ because we are considering a problem of fixed-size subset selection. When we look at all possible subsets of size $k$, the term $\alpha_0 \lambda_0 k n g_{nm}$ turns out to be a constant value for all subsets of size $k$ and does not make any difference during the submodular maximization across different subsets. I hope this answers your question. Please let us know if you have any more questions regarding the work. We would be happy to help.

---

> > ### Public Comment · ~Xilie_Xu1 · 2022-12-06
> > **Thanks for response**
> >
> > Hi Krishnateja, many thanks for your clarification. I got it!

---

### Decision · Program_Chairs · 2021-09-27

**Decision:**

Accept (Poster)

**Comment:**

This paper proposes RETRIEVE to address the computational cost issue in previous semi-supervised learning algorithms. It is well-written and well-motivated. The proposed idea is incremental but technically sound. The claims are well supported by theoretical analyses and extensive experimental results.